# The E3 ubiquitin ligase Cul4b promotes CD4+ T cell expansion by aiding the repair of damaged DNA

Asif A. Dar[1], Keisuke Sawada[1], Joseph M. Dybas[1,2], Emily K. Moser[1], Emma L. Lewis[3], Eddie Park[4], Hossein Fazelinia[5], Lynn A. Spruce[5], Hua Ding[5], Steven H. Seeholzer[5], Paula M. Oliver[1,6]*

1 Division of Protective Immunity, Children's Hospital of Philadelphia, Philadelphia, Pennsylvania, United States of America, 2 Department of Biomedical Health and Informatics, Children's Hospital of Philadelphia, Philadelphia, Pennsylvania, United States of America, 3 Medical Scientist Training Program, Perelman School of Medicine, University of Pennsylvania, Philadelphia, Pennsylvania, United States of America, 4 Center for Computational and Genomic Medicine, The Children's Hospital of Philadelphia, Philadelphia, Pennsylvania, United States of America, 5 Division of Cell Pathology, Children's Hospital of Philadelphia, Philadelphia, Pennsylvania, United States of America, 6 Department of Pathology, University of Pennsylvania, Philadelphia, Pennsylvania, United States of America

* paulao@pennmedicine.upenn.edu

**Data Availability Statement:** All the relevant data are noted within the paper and included as supporting information files. The mass spectrometry proteomics data have been deposited

## Abstract

The capacity for T cells to become activated and clonally expand during pathogen invasion is pivotal for protective immunity. Our understanding of how T cell receptor (TCR) signaling prepares cells for this rapid expansion remains limited. Here we provide evidence that the E3 ubiquitin ligase Cullin-4b (Cul4b) regulates this process. The abundance of total and neddylated Cul4b increased following TCR stimulation. Disruption of Cul4b resulted in impaired proliferation and survival of activated T cells. Additionally, Cul4b-deficient CD4+ T cells accumulated DNA damage. In T cells, Cul4b preferentially associated with the substrate receptor DCAF1, and Cul4b and DCAF1 were found to interact with proteins that promote the sensing or repair of damaged DNA. While Cul4b-deficient CD4+ T cells showed evidence of DNA damage sensing, downstream phosphorylation of SMC1A did not occur. These findings reveal an essential role for Cul4b in promoting the repair of damaged DNA to allow survival and expansion of activated T cells.

## Introduction

One of the most fundamental elements of adaptive immunity is the clonal expansion of pathogen-specific T cells [1,2]. The T cell's ability to rapidly multiply to generate cell numbers sufficient to control viral and certain bacterial infections can prove crucial for the host's survival. Thus, T cells must be wired to undergo cell division at an exceptionally rapid rate [3,4]. Indeed, activated T cells proliferate at one of the most rapid rates of all mammalian cell types, matched only by cells in certain embryonic tissues [5]. Such a rapid rate of proliferation requires extremely rapid DNA synthesis and short cell cycle times.

to the ProteomeXchange Consortium via the PRIDE partner repository with the dataset identifier PXD017699 and PXD019272. All FCS files are available on flowrepository platform (https://flowrepository.org/) using ID: FR-FCM-Z38M.

**Funding:** The author(s) received no specific funding for this work.

**Competing interests:** The authors have declared that no competing interests exist.

**Abbreviations:** BM, bone marrow; CBF, chromatin-bound fraction; CD, cluster of differentiation; CF, cytoplasmic; fraction; CFSE, carboxyfluorescein succinimidyl ester; CHX, cycloheximide; CRL, Cullin RING Ligase; Cul4b, Cullin-4b; DCAFs, DDB1- and CUL4-associated factors; DDB1, damaged DNA-binding protein 1; DDR, DNA damage response; DN, double negative; DP, double positive; HSPC, hematopoietic stem and progenitor cell; iBAQ, intensity-based absolute quantification; IL-2, interleukin 2; IL-2R, IL-2 receptor; IP, immunoprecipitation; MACS, magnetic-activated cell sorting; MRN, MRE11–RAD50–NBS1; MS, mass spectrometry; MS/MS, tandem mass spectrometry; NBS, Nijmegen breakage syndrome; NF, nuclear-soluble fraction; o-PA, ortho-phenanthroline; PI, propidium iodide; pSMC1A, SMC1A phosphorylation; RP-HPLC, Reverse Phase-High Performance Liquid Chromatographic; SEM, standard error of mean; SP, single positive; TCR, T cell receptor; VprBp, Vpr-binding protein; WB, western blotting; WT, wild-type; XLMR, X-linked mental retardation.

In T cells, as in other mammalian cell types, cell division is controlled at multiple points throughout the cell cycle [6,7]. One of the prominent regulatory checkpoints is the transition from $G_1$ to S phase and the onset of DNA synthesis [8,9]. In T cells, the need for an exceedingly rapid rate of DNA synthesis is thought to be achieved by initiating multiple origins of replication within the DNA [5]. In the absence of effective DNA repair mechanisms, activated T cells would be highly susceptible to replication induced genomic stress and DNA damage [10,11]. Unrepaired DNA damage could induce permanent cell cycle arrest or apoptosis, and culling of the very cells needed to control the invading pathogen. However, mechanisms that allow T cells to rapidly repair DNA during clonal expansion are poorly understood.

One mechanism that promotes DNA repair and cell division in rapidly dividing tumor cells relies on the Cullin RING Ligase (CRL) CRL4 (reviewed in [12]). There are 2 proteins that provide a docking platform for CRL4 E3 ubiquitin ligase formation; Cul4a and Cul4b. While highly homologous, these 2 proteins are encoded on different chromosomes [13]. Cul4a and Cul4b are both activated by neddylation and both interact with Damaged DNA binding protein 1 (DDB1), an adaptor that recruits substrate receptors to the Cul4 complex [14,15]. Given their high homology and usage of the same adaptor protein, Cul4a and Cul4b were initially thought to have mostly redundant functions. However, studies have revealed both distinct and overlapping functions [13]. Supporting that these 2 proteins are not redundant, loss-of-function mutations in Cul4b, but not Cul4a, have been identified in patients with cerebral cortical malformations and X-linked mental retardation (XLMR) [16–19]. However, both Cul4a and Cul4b can promote cancer. Elevated expression of these proteins has been observed in many types of tumors, and high expression correlates with alterations in tumor suppressor and cell cycle regulator genes [20–22]. In cancer cells, Cul4a and Cul4b have been shown to regulate proliferation, DNA damage and repair, cell cycle progression, DNA methylation and histone acetylation, as well as cell signaling [23–27]. Whether Cul4a or Cul4b regulate the proliferation of T cells is not known.

We found that Cul4b was expressed at low levels in naïve T cells but was increased in abundance and activation following T cell receptor (TCR) ligation. Furthermore, Cul4b was significantly more abundant than Cul4a in both naïve and activated CD4+ T cells. To assess the role of Cul4b in T cells, we generated Cul4b^fl/fl CD4-Cre mice. Loss of Cul4b resulted in a decreased frequency of activated (CD44^hi) T cells under competitive environment in vivo. Additionally, Cul4b was required for the expansion and pathogenicity of T cells. Coculture experiments revealed that loss of Cul4b resulted in reduced proliferation and survival of TCR-activated CD4+ T cells. Proteomics analysis of Cul4b-deficient CD4+ T cells revealed an alteration in cell cycle and DNA damage-related pathways. Cul4b-deficient T cells acquired more DNA damage and had higher expression of DNA damage recognition proteins. In activated T cells, Cul4b interacted most abundantly with Vprbp (DCAF1), and both Cul4b and DCAF1 were found to associate with MRE11A, RAD50, and SMC1A, proteins associated with DNA damage response and repair. Cul4b deletion also resulted in attenuated ubiquitination and phosphorylation of SMC1A without impacting its levels or recruitment to chromatin. Together these findings demonstrate that Cul4b allows the expansion of activated T cells by regulating repair of damaged DNA.

## Results

### T cell activation increases the abundance of total and neddylated Cul4a and Cul4b

We initially identified Cul4b in a screen that used di-glycine remnant profiling [28,29] in a manner that revealed cullin ligases that were particularly active (neddylated) in T cells

following TCR ligation [30]. The covalent attachment of Nedd8 ("neddylation") of a cullin E3 ligase induces a conformational change in the CRL complex that increases its enzymatic activity [31]. Such neddylation can be blocked using the Nedd8-activating enzyme inhibitor MLN4924 [32]. Since a covalently bound Nedd8 generates a di-glycine motif upon trypsinization, neddylation of a cullin ligase can be detected using di-glycine remnant profiling in the presence or absence of MLN4924 [30]. Furthermore, neddylation of a substrate may be identified by a shift to a higher molecular weight band (approximately 8 kDa) in traditional western blotting (WB) of protein lysates. In our initial study, the di-glycine remnant peptide of Cul4b was only identified after T cell activation, supporting that Cul4b neddylation was more abundant following TCR ligation. Furthermore, we did not identify a di-glycine remnant for Cul4a, suggesting that Cul4a was of low abundance or not neddylated under the same conditions.

To assess the expression patterns of Cul4a and Cul4b in T cells, we purified naive CD4$^+$ and CD8$^+$ T cells from wild-type (WT) mice and stimulated for various time points with anti-CD3 and anti-CD28. We found that protein levels of Cul4a and Cul4b were low in naive T cells, but levels were significantly increased following TCR stimulation (Fig 1A). Expression and neddylation of both proteins was increased by 24 h after stimulation and then remained elevated over levels observed in naïve CD4$^+$ and CD8$^+$ T cells (Fig 1B–1D, S1A–S1C Fig). Both neddylated (active) and nonneddylated (inactive) forms were observed for Cul4a and Cul4b following TCR stimulation. Cul4b was predominantly present in its neddylated (active) form in both CD4$^+$ and CD8$^+$ T cells (Fig 1C, S1B Fig), whereas Cul4a had a greater proportion in its nonneddylated (inactive) state compared to Cul4b (Fig 1D, S1C Fig). When we assessed the relative ratios of neddylated and nonneddylated forms at different time points, we found a higher fraction of Cul4b was present in its active form (Fig 1E and 1F). Neddylation and deneddylation are highly sensitive processes, and Nedd8 can be removed from Cul4 during incubation with lysis buffers that fail to inactivate enzymatic activity (S1D Fig). Thus, to maintain the neddylation state of Cul4b after cell lysis, these experiments were performed by lysing cells in SDS sample buffer (Fig 1A, S1D Fig). Supporting that the slower migrating band was indeed neddylated Cul4, addition of MLN4924 triggered a rapid loss of this band in both the Cul4a and Cul4b blots (Fig 1A). These results revealed that both Cul4a and Cul4b are expressed in murine T cells and that their expression is increased following T cell activation.

## Cul4b is more abundant than Cul4a in CD4$^+$ T cells

It is well established that T cells require both TCR and costimulatory signals to become fully activated (reviewed in [33]). To address, whether one or both of these signals was required for the increase in Cul4 protein levels, we purified naive CD4$^+$ T cells from WT mice and stimulated them either with anti-CD3 alone or together with anti-CD28. We found that anti-CD3 alone did not alter the levels of Cul4a or Cul4b, while stimulation with anti-CD3/28 resulted in higher levels of both proteins (Fig 2A). Anti-CD3/28 activation of T cells promotes interleukin 2 (IL-2) secretion [34] and autocrine IL-2 receptor (IL-2R) signaling. To determine whether the addition of anti-CD28 or IL-2R signaling was responsible for the increased Cul4 levels, we pharmacologically blocked the IL2Rα receptor. IL-2R blockade had no notable impact on the anti-CD3/28-mediated increase of Cul4a and Cul4b (Fig 2A). To determine whether the increase in Cul4 protein levels was due to increased transcription, we assessed Cul4b transcripts by quantitative PCR. The data confirmed that Cul4b transcript levels were not detectable in naïve or anti-CD3 alone stimulated T cells. However, stimulation of CD4$^+$ T cells with anti-CD3/CD28 resulted in the robust increase in Cul4b transcripts, independent of IL-2 receptor signaling (S2A Fig). These results supported that TCR ligation, together with CD28 costimulation, drives an increase in the levels of Cul4 proteins.

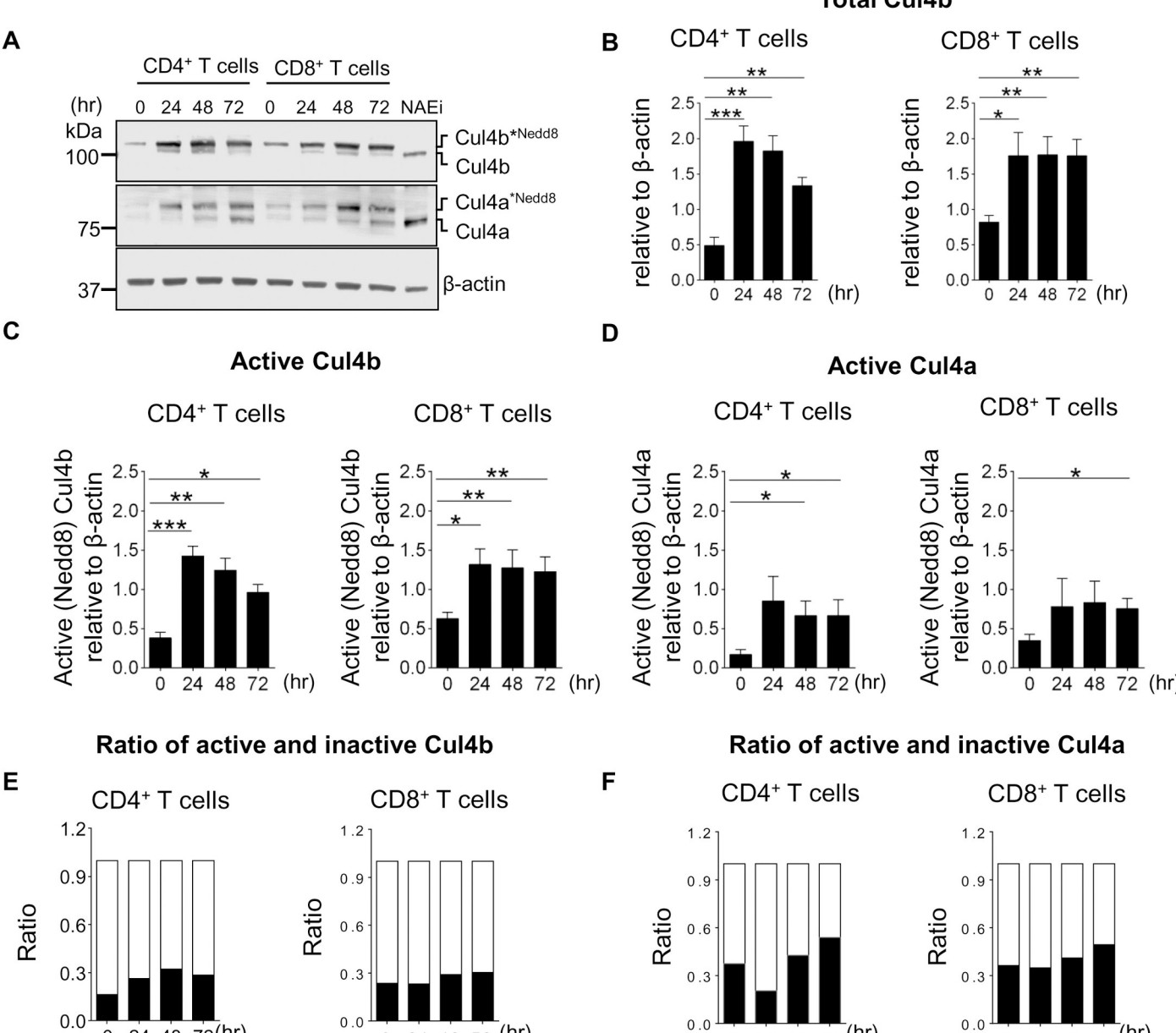

**Fig 1. TCR-driven activation of Cul4a and Cul4b.** Naive CD4+ T and CD8+ T cells from control mice (C57BL/6 mice) were activated by anti-CD3 and anti-CD28 mAbs. At indicated time points after activation, cell lysates were prepared, and expression of Cul4a and Cul4b was monitored by immunoblotting. (**A**) Immunoblot for Cul4a and Cul4b shows presence of neddylated and nonneddylated forms of the protein. Treatment of activated CD4+ T cells with NAEi (1 μM for last 1 h of culture) removed the neddylation from both the proteins. Data across the lanes were normalized with β-actin. (**B–D**) The quantitative data of 4 independent experiments are shown. (**B**) shows total Cul4b (neddylated and nonneddylated) in CD4+ and CD8+ T cells at different time points. (**C**) Active (neddylated) Cul4b in CD4+ and CD8+ T cells. (**D**) Active Cul4a in CD4+ and CD8+ T cells. (**E and F**) Shows the relative ratios of neddylated and nonneddylated forms of Cul4b and Cul4a at different time points in CD4+ and CD8+ T cells. Data were quantitated using Image J software and is represented as mean ± SEM (*P < 0.05 **P < 0.01, ***P < 0.001 by Student t test). For numerical raw data, please see S1 Data. Cul4a, Cullin-4a; Cul4b, Cullin-4b; mAbs, monoclonal antibodies; NAEi, Nedd8 activating enzyme inhibitor; SEM, standard error of mean; TCR, T cell receptor.

To assess the subcellular localization of Cul4a and Cul4b in CD4+ T cells, we analyzed their relative abundances in cytoplasmic and nuclear fractions. CD4+ T cells were stimulated for 48 h, and cytoplasmic (CF), nuclear-soluble (NF), and chromatin-bound (CBF) fractions were

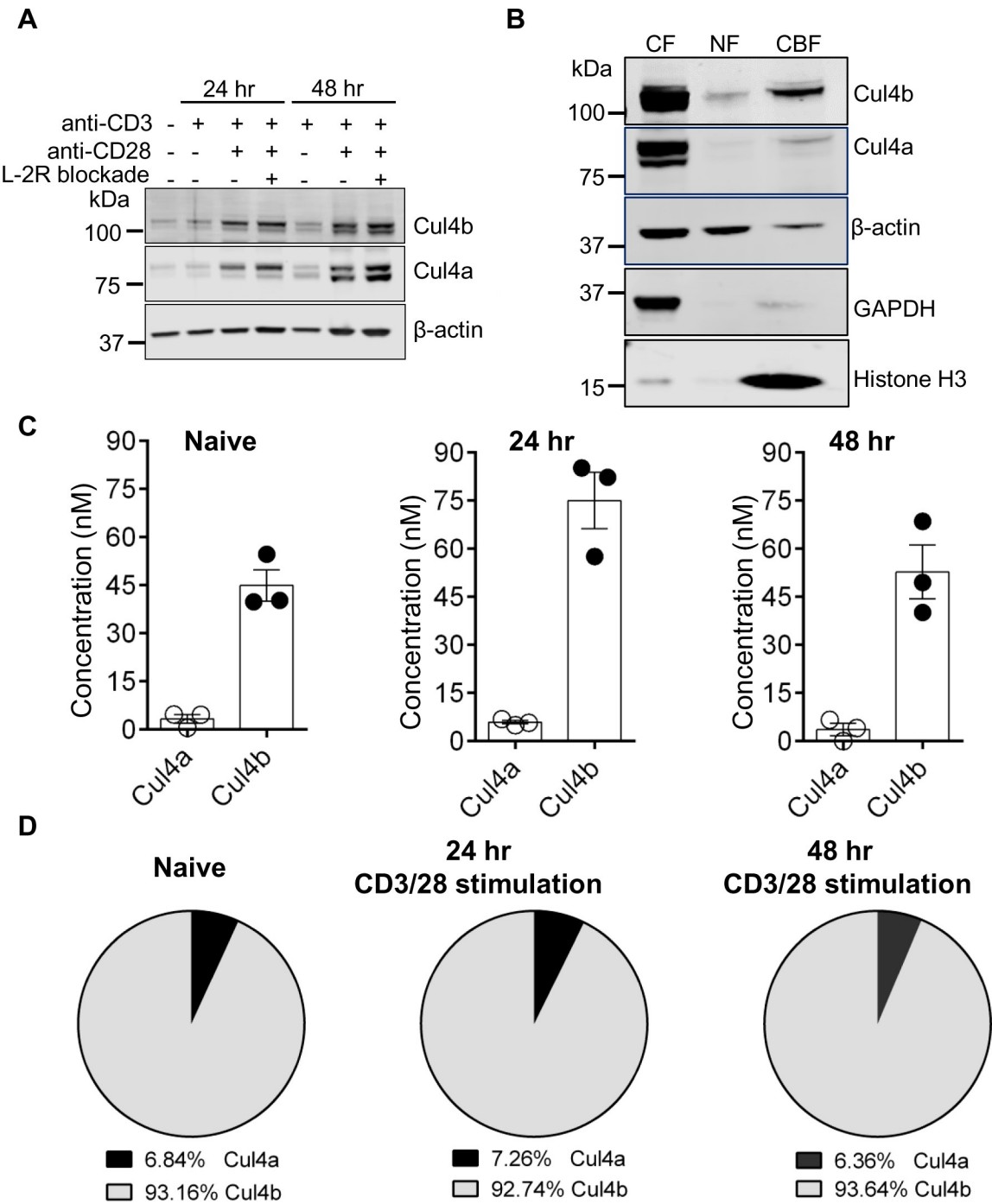

**Fig 2. Cul4b is dynamically regulated in CD4+ T cells. (A)** Naive CD4+ T cells from control mice were activated either with anti-CD3 mAb alone or anti-CD3 and anti-CD28 mAbs. In case of anti-CD3/CD28 mAb stimulation, cells were either untreated or neutralized for IL-2R by adding anti-IL-2R antibody (10 μg/ml). At indicated time points after stimulation, cell lysates were prepared, and expression of Cul4a and Cul4b was monitored by immunoblotting. **(B)** CD4+ T cells were stimulated for 48 h, and cytoplasmic (CF), nuclear-soluble (NF), and chromatin-bound (CBF) proteins were harvested and analyzed by immunoblot. Histone H3 shows enrichment of CBF, and GAPDH shows enrichment of CF. β-actin shows the amount of protein used for CF and NF. **(C)** Quantification of protein abundance of Cul4a and Cul4b in CD4+ T cells in naïve and TCR-activated (24 and 48 h) CD4+ T cells is shown and was calculated using the proteomic ruler method [37]. **(D)** The pie chart shows the relative proportion of Cul4a and Cul4b in naive and activated T cells. The percentages are calculated from the mean concentrations of Cul4a and Cul4b from 3 different samples. For numerical raw data, please see S2 Data. CBF, chromatin-bound fraction; CF, cytoplasmic fraction; Cul4a, Cullin-4a; Cul4b, Cullin-4b; GAPDH, glyceraldehyde 3-phosphate dehydrogenase; mAbs, monoclonal antibodies; NF, nucleoplasmic fraction; TCR, T cell receptor.

obtained. Bands corresponding to both neddylated and unneddylated forms Cul4b and Cul4a were found in the cytoplasm (Fig 2B). While there was little Cul4a observed in nuclear fractions, Cul4b was localized within the nuclei, predominantly in the CBF (Fig 2B). Interestingly, only a band corresponding to the neddylated form of Cul4b was seen in the CBF. The finding that Cul4b exists as a nuclear protein is consistent with the other previous reports [35,36] and implies that Cul4b function in T cells may involve its association with chromatin.

Given our observation that both Cul4a and Cul4b were expressed in T cells, we next examined the relative abundance of Cul4a and Cul4b using mass spectrometry. Purified naïve CD4$^+$ T cells were stimulated with anti-CD3/CD28 for 24 and 48 h, and abundances were determined by calculating the concentration and copy numbers using mass spectrometry. The absolute protein copies per cell were calculated using the "proteomic ruler" method, which uses the mass spectrometry signal of histones as an internal standard [37]. This method avoids error-prone steps of cell counting and protein concentration evaluation and can be used to estimate protein abundance per cell. At each time point, the concentration and copy numbers of Cul4b were more than Cul4a (Fig 2C, S2B Fig). Cul4b represented the dominant form and constituted about 95% of total Cul4 proteins in CD4$^+$ T cells (Fig 2D). To confirm this, we reanalyzed the dataset reported by Howden and colleagues [38]. In this dataset, we used intensities to assess the copy number of Cul4a and Cul4b. Cul4b had a higher copy number and was found at a higher concentration than Cul4a, both in naive and antigen-stimulated CD4$^+$ T cells (S2C Fig). Taken together, these results support that Cul4b is much more abundant than Cul4a in activated CD4$^+$ T cells.

## Cul4b regulates the expansion and pathogenicity of CD4$^+$ T cells in a model of colitis

To study the function of Cul4b in T cells, we generated Cul4b-floxed (Cul4b$^{fl/fl}$) mice using CRISPR-Cas9 technology. LoxP sites were introduced on either side of exon 4 (Fig 3A). Cre-mediated recombination of the 2 LoxP sites resulted in a deletion of exon 4 and a frameshift upon aberrant splicing of exons 3 and 4. To delete Cul4b specifically in mature T cells, Cul4b$^{fl/fl}$ mice were crossed with CD4Cre mice (Fig 3A). To assess the efficiency of deletion of Cul4b, CD4$^+$ T cells were isolated from spleens of Cul4b$^{fl/fl}$CD4Cre mice, and Cul4b was analyzed by WB. We found that Cul4b was effectively deleted in CD4$^+$ T cells and this did not impact the expression of Cul4a (Fig 3B).

Cul4b$^{fl/fl}$CD4Cre mice were born at the predicted mendelian frequencies and were phenotypically normal. T cell development was largely normal in Cul4b$^{fl/fl}$CD4Cre mice. Compared with Control (Cul4b$^{fl/fl}$) littermates, Cul4b$^{fl/fl}$CD4Cre mice had similar percentages of double negative (DN), double positive (DP), and CD4 single positive (SP), but slightly fewer CD8SP thymocytes (Fig 3C and 3D). The numbers of CD4SP and CD8SP cells in thymus were similar (S3A Fig). The percentages and numbers of mature CD4$^+$ T cells in the peripheral lymphoid organs (spleen and lymph nodes) were also similar (Fig 3E and 3F, S3B–S3E Fig), although CD8$^+$ T cells were somewhat lower in Cul4b$^{fl/fl}$CD4Cre than in control (Cul4b$^{fl/fl}$) mice (Fig 3E and 3F,S3B–S3E Fig). Additionally, there were similar frequencies of naïve (CD44$^{lo}$CD62L$^{hi}$) and effector (CD44$^{hi}$CD62L$^{lo}$) CD4$^+$ and CD8$^+$ T cells (Fig 3G).

Given that expression of Cul4b increased following TCR stimulation, we posited that it might be particularly important in activated T cells. To test this hypothesis, we adoptively transferred naïve CD4$^+$ T cells from control (Cul4b$^{fl/fl}$ (WT)) and Cul4b$^{fl/fl}$CD4Cre (KO) mice into Rag1$^{-/-}$ recipients. In this T cell-mediated model of colitis, the transfer of naïve CD4$^+$ T cells into lymphopaenic recipients (Rag1$^{-/-}$) results in colitis that is driven by the activation and expansion of the transferred T cells. Once activated, T cells expand, infiltrate the colon,

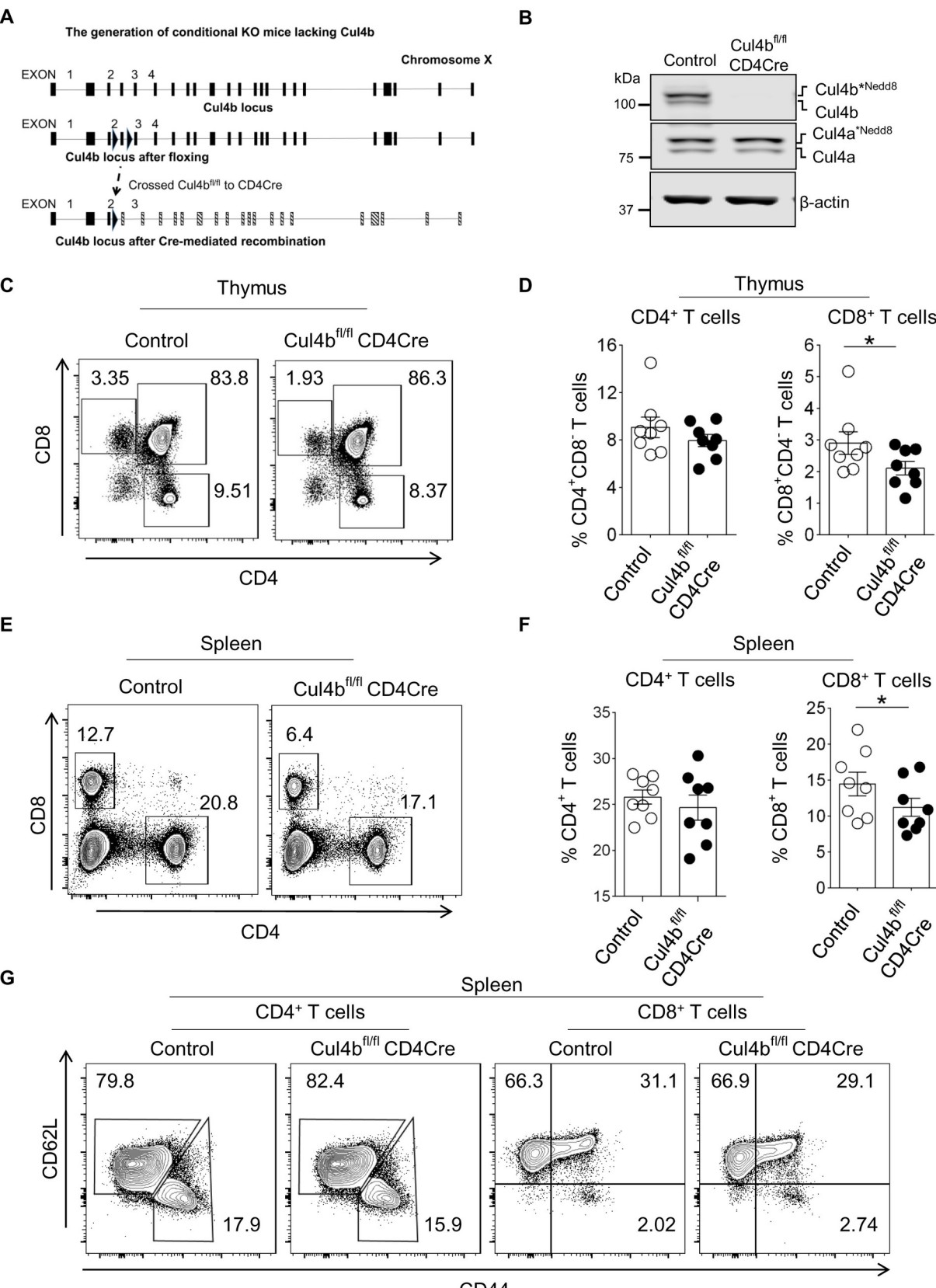

**Fig 3. T Cul4b deletion is dispensable for mature cells. (A)**. Cul4b-floxed alleles were generated using the CRISPR/Cas9 technology. Exons 3 and 4 were flanked by 2 loxP sites. Mouse with floxed Cul4b alleles were bred with CD4-Cre mice. The Cre recombinase catalyzes the recombination between 2 LoxP sites to disrupt the reading frame of Cul4b mRNA. **(B)** Immunoblot showing expression of Cul4b and Cul4a in sorted CD4$^+$ T cells isolated from Control (Cul4bfl/fl) and conditional knockout mice (Cul4b$^{fl/fl}$-CD4Cre). β-actin was used as internal loading control. **(C–F)** The distribution of various T-cell populations in the thymus and spleen of control (Cul4b$^{fl/fl}$) and Cul4b$^{fl/fl}$-CD4Cre mice was assessed by flow cytometry. The bar graphs show the mean ± SEM of 8 sets of mice (*$P < 0.05$ by Student $t$ test ns, not significant, $P > 0.05$ by Student $t$ test). The mice were paired with respective to age, gender, cage, and time of takedown. **(G)** The distribution of naive (CD62L$^{high}$CD44$^{low}$) CD4$^+$ and CD8$^+$ T cells, effector/memory (CD62L$^{low}$CD44$^{high}$) CD4$^+$ and effector/memory (CD62L$^+$CD44$^{high}$) CD8$^+$ T cells in the spleen of mice with different genotypes, assessed by flow cytometry. For numerical raw data, please see S3 Data. Cul4a, Cullin-4a; Cul4b, Cullin-4b; SEM, standard error of mean.

and drive inflammation and tissue destruction [39]. Consistent with previous reports [39], transfer of WT (control) naïve T cells into Rag1$^{−/−}$ recipients resulted in T cell expansion and colitis. While mice that received control (Cul4b$^{fl/fl}$) CD4$^+$ T cells developed a progressive weight loss starting at 3 weeks post transfer, mice which received Cul4b$^{fl/fl}$CD4Cre T cells showed little to no weight loss (Fig 4A). This supported that Cul4b$^{fl/fl}$CD4Cre cells were less colitogenic. Thus, we analyzed the numbers and percentages of CD4$^+$ T cells in spleen and colon of the recipient mice. We found significantly fewer T cells in animals that received Cul4b$^{fl/fl}$CD4Cre cells compared to those that received control cells (Fig 4B and 4C). Additionally, animals that received control (Cul4b$^{fl/fl}$) cells showed severe inflammation of the colon 8 weeks after cell transfer, while mice that received Cul4b$^{fl/fl}$CD4Cre cells showed no or only modest signs of inflammation (Fig 4D). To determine whether loss of Cul4b impacted T cell proliferation in vivo, we examined the levels of Ki67, a protein expressed in cells that have recently divided [40]. The frequency of Ki67-positive cells among Cul4b$^{fl/fl}$CD4Cre CD4$^+$ T cells was lower than that of control CD4$^+$ T cells in both spleen and colon. These data suggested that Cul4b was required for the expansion of CD4$^+$ T cells in secondary lymphoid compartments (Fig 4E and 4F). Taken together, these data indicated that Cul4b was required for the expansion and pathogenicity of activated T cells.

## Cul4b promotes the maintenance of CD4$^+$ effector (CD44$^{hi}$CD62L$^{lo}$) T cell numbers

Having determined that Cul4b was crucial for CD4$^+$ T cell expansion, and given the competition that exists among T cells as they populate peripheral niches [41,42], we posited that Cul4b-deficient T cells would be ineffective competitors if matched with control cells. To test the fitness of Cul4b$^{fl/fl}$CD4Cre T cells in a competitive environment, we generated mixed-bone-marrow (BM) chimeras. To do this, we injected congenically distinct BM cells (T cell depleted) from control (CD45.1$^+$) and Cul4b$^{fl/fl}$CD4Cre (CD45.2$^+$) mice into sublethally irradiated Rag1$^{−/−}$ mice. T cells were then analyzed after 8 to 10 weeks after BM cell transfer. We normalized congenically distinct T cell populations to their B cell counterparts, since B cell numbers would be identical with respect to Cul4b expression. When analyzing overall frequencies of CD4$^+$ T cells, we found significantly fewer Cul4b-deficient cells compared to controls (S4A–S4C Fig). Given that Cul4b expression is higher in activated T cells, we analyzed naïve (CD44$^{lo}$CD62L$^{hi}$) and effector/activated (CD44$^{hi}$CD62L$^{lo}$) T cells separately to determine whether these populations were similarly or disparately impacted. We found that while naïve CD4$^+$ cells were found at comparable frequencies, effector CD4$^+$ T cells were significantly less frequent than their Cul4b expressing controls (Fig 5A–5F). Furthermore, Cul4b$^{fl/fl}$CD4Cre T cell subsets exhibited significantly fewer Ki67$^+$ cells, supporting a decreased proliferative capability of these cells (S4D–S4G Fig). Altogether, this confirms that effector CD4$^+$ T cells that lacked Cul4b were poorly maintained in the periphery when in competition with

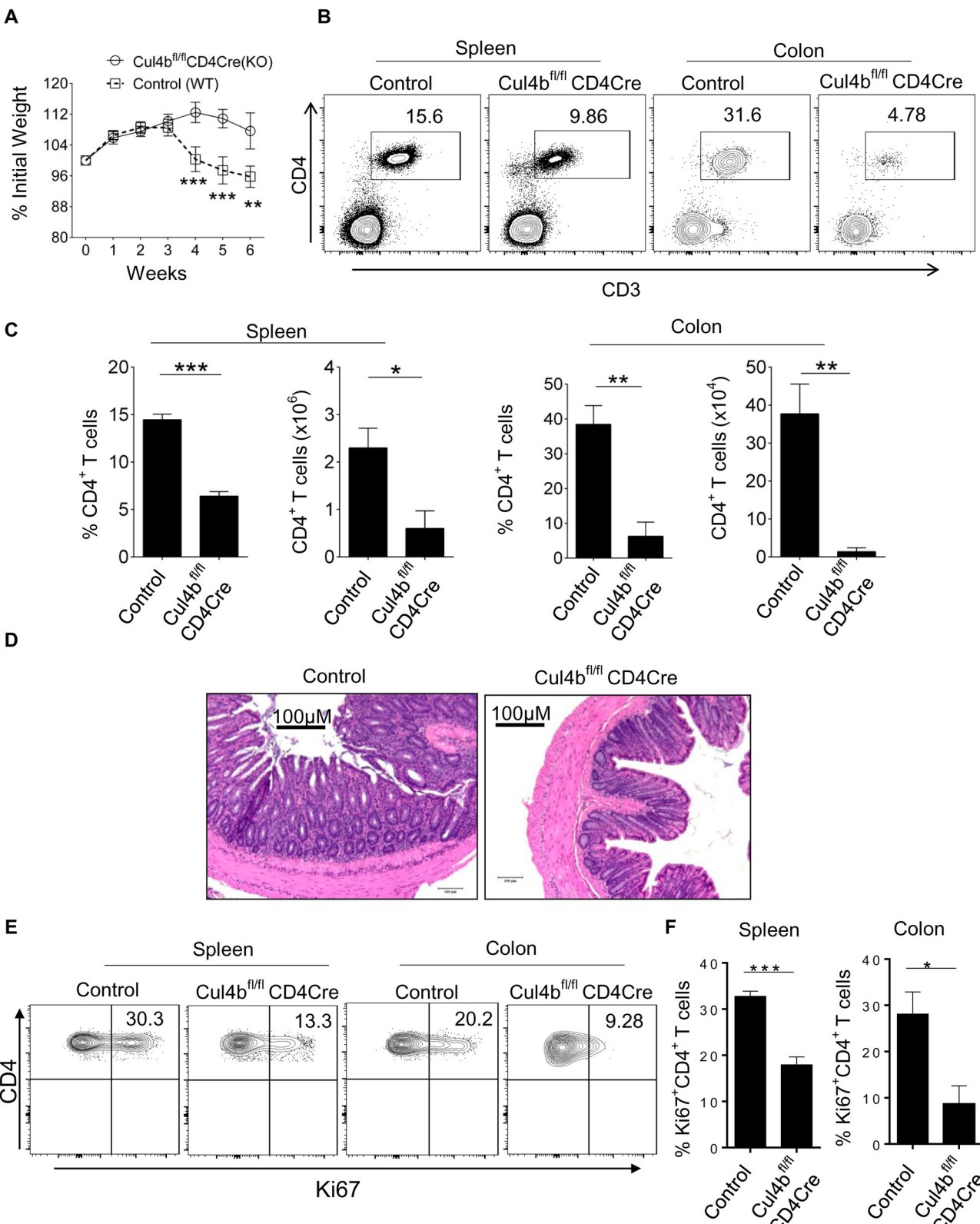

**Fig 4. Cul4b regulates pathogenicity of CD4+ T cells.** Each Rag1−/− recipient mice was adoptively transferred with 1x10^6 CD4+CD25−CD44−CD62L^hi naive T cells isolated from control and Cul4b^fl/fl-CD4Cre mice. (**A**) The weight loss of the mice was monitored every 7 days for 6–7 weeks from the day after intraperitoneal injection. *n* = 5, recipient mice for each group were used and experiment was repeated twice. (**B**) The distribution of CD4+ T cell

populations in the spleen and colon of Rag1$^{-/-}$ mice which received control (Cul4b$^{fl/fl}$) and Cul4b$^{fl/fl}$-CD4Cre CD4$^+$ T cells was assessed by flow cytometry. (**C**) The bar graphs show the percentage and numbers of CD4$^+$ T cells in the spleen and colon of Rag1$^{-/-}$ recipient. Data are represented as mean ± SEM and are representative of one of the 2 experiments ($n = 5$) (*$P < 0.05$ **$P < 0.01$, ***$P < 0.001$ by Student $t$ test). (**D**) Representative H&E staining of colon sections from Rag1$^{-/-}$ mice transferred with control naïve CD4$^+$ T cells or Cul4b$^{fl/fl}$-CD4Cre naïve CD4$^+$ T cells, (magnification 10×, scale bar 100 μM). (**E and F**) CD4$^+$ T cells in spleen and colon were analyzed for Ki67 expression. Representative plots show the frequencies of Ki67-positive cells in recipients that received control or Cul4b$^{fl/fl}$-CD4Cre CD4$^+$ T cells. The bar graphs show the relative frequencies of Ki67$^+$CD4$^+$ T cells in spleen and colon. (*$P < 0.05$, ***$P < 0.001$ by Student $t$ test). For numerical raw data, please see S4 Data. Cul4b, Cullin-4b; SEM, standard error of mean.

Cul4b-expressing CD4$^+$ T cells. These findings thus demonstrated that Cul4b promotes the maintenance of effector CD4$^+$ T cells.

## Cul4b is required for the proliferation and survival of CD4$^+$ T cells

Given that Cul4b was up-regulated following T cell stimulation (Fig 1A and 1B) and that Cul4b was required for effector T cell maintenance (Fig 5D–5F), we sought to determine whether Cul4b might promote T cell activation, survival, or proliferation. To test this, we cocultured naïve CD4$^+$ T cells isolated from control (CD45.1$^+$) and Cul4b$^{fl/fl}$CD4Cre (CD45.2$^+$) mice in the presence of anti-CD3 and anti-CD28. The up-regulation of CD69, CD25, and CD44 was similar between control and Cul4b-deficient T cells (S4H Fig), suggesting that Cul4b did not impact the ability of T cells to be activated. However, after 5 days of coculture, the frequency of Cul4b-deficient cells was diminished (Fig 5G), and higher numbers of control cells were observed (Fig 5H). In keeping with this, Cul4b-deficient CD4$^+$ T cells proliferated less than control cells as evidenced by loss of carboxyfluorescein succinimidyl ester (CFSE) (Fig 5I), and Cul4b-sufficient cells showed a higher division index (Fig 5J). The reduced proliferation in Cul4b-deficient CD4$^+$ T cells could result from their poor survival rate. To test this, we stimulated cells under in vitro culture conditions and analyzed cell death using Annexin V staining. We found significantly higher percentages of Cul4b-deficient CD4$^+$ T cells undergoing apoptosis (Fig 5K and 5L). These data support that, once activated, Cul4b-deficient CD4$^+$ T cells were less likely to proliferate and more prone to apoptosis than their control counterparts.

## Cul4b deletion increases DNA damage in activated CD4$^+$ T cells

To identify the biological processes regulated by Cul4b in activated CD4$^+$ T cells, we used quantitative mass spectrometry to profile the proteomes of control and Cul4b-deficient CD4$^+$ T cells following anti-CD3/CD28 activation. To quantify changes in protein expression, we compared protein abundances in control and Cul4b-deficient CD4$^+$ T cells using label-free intensity-based absolute quantification (iBAQ) [43]. More than 6,400 proteins were quantified, with 567 proteins being significantly ($P < 0.05$) increased or decreased in abundance in Cul4b-deficient CD4$^+$ T cells (S5A Fig, S1 Table). Significantly up- and down-regulated proteins were functionally annotated using DAVID bioinformatics tool. Ontological and functional analysis of the 567 proteins was carried out using GOplot algorithm and is depicted in the GOcircular plot (Fig 6A). In this plot, the outer circle is a scatter plot for each biological term of the logFC (fold change) of the enriched proteins. Red dots show the up-regulated proteins in the Cul4b-deficient CD4$^+$ T cells and blue dots down-regulated proteins. We found that the most enriched protein clusters were those associated with cell cycle and the DNA damage response (Fig 6A). Considering that DNA damage can result in cell cycle arrest and apoptosis, we next sought to test the ability of Cul4b-deficient cells to progress through the cell cycle and their ability to sense DNA damage. To test this, we stimulated cells under in vitro culture conditions and analyzed cell cycle using propidium iodide (PI). We found that, in the

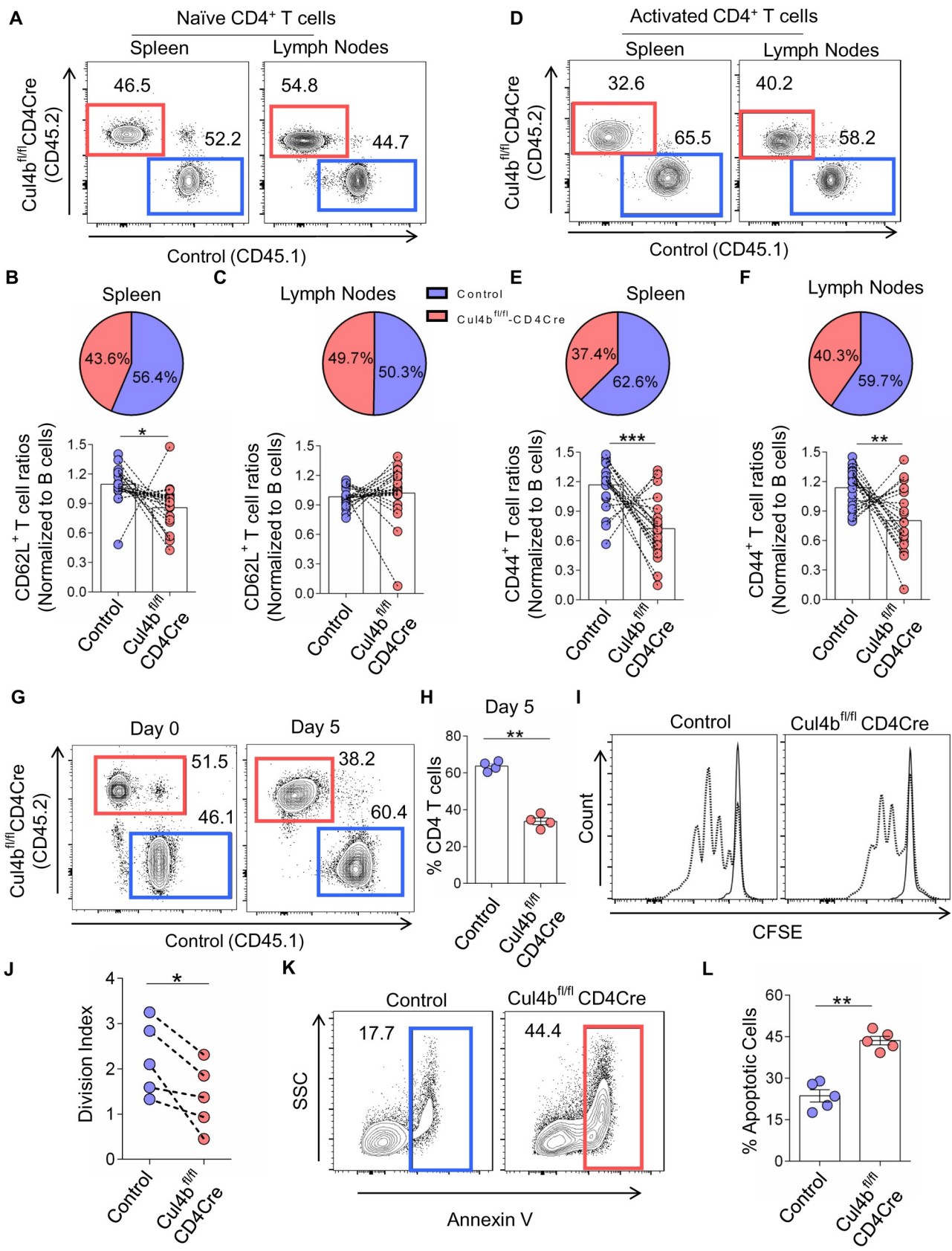

**Fig 5. Cul4b regulates homeostasis, proliferation, and survival of activated CD4⁺ T cells.** T cell-depleted BM cells were injected into sublethally irradiated $Rag1^{-/-}$ recipient mice. Congenically marked BM cells from Cul4bfl/fl-CD4Cre (CD45.2) and control (CD45.1) were mixed into 1:1 ratio, and $2 \times 10^6$ cells were injected into each recipient mouse. **(A)** The comparison of the naïve (CD62L$^{hi}$CD44$^{lo}$) CD4⁺ populations in the spleen and lymph nodes of irradiated recipient chimeric mice after reconstitution of BM cells from control mice (CD45.1⁺) and Cul4b$^{fl/fl}$-CD4Cre mice (CD45.2⁺). **(B and C)** The line graphs show the relative ratios of CD62L$^{hi}$CD44$^{lo}$ CD4⁺ T cells. Ratios were calculated by dividing the percentages of naïve T cells of each genotype with the percentages of B cells from the same genotype. The pie chart depicts the relative percentages of the control and Cul4b$^{fl/fl}$-CD4Cre CD62L$^{hi}$CD44$^{lo}$ CD4⁺ T cells **(D–F)**. The comparison of activated/effector memory (CD62L$^{lo}$CD44$^{high}$) CD4⁺ T cells in the spleens and lymph nodes of the recipient chimeric mice. The line graphs show the relative ratio of CD62L$^{lo}$CD44$^{high}$ T cells calculated by dividing the percentages of activated T cells with the percentages of B cells from the respective genotypes. The pie chart depicts the relative percentages of the control and Cul4b$^{fl/fl}$-CD4Cre CD62L$^{lo}$CD44$^{hi}$ CD4⁺ T cells. Data from 20 recipient mice are shown; experiment was repeated twice with 2 donors in each experiment ($n = 5$, recipients for each donor ($n = 4$)) (*$P < 0.05$ **$P < 0.01$, ***$P < 0.001$ by Student $t$ test). **(G)** Naïve CD4⁺ T cells isolated from control mice (CD45.1⁺) and Cul4b$^{fl/fl}$-CD4Cre mice (CD45.2 mice were cocultured and stimulated in vitro with anti-CD3 and CD28 mAbs (5 μg/ml) for 5 days. The relative proportions of the cells were assessed by flow cytometry. **(H)** The bar graph shows the mean ± SEM percentages of cells of 4 independent experiments ($n = 4$). **$P < 0.01$ by Student $t$ test. **(I)** Naïve CD4⁺ T cells isolated from control (CD45.1⁺) and Cul4b$^{fl/fl}$-CD4Cre (CD45.2⁺) mice were labeled with CFSE, then cocultured and stimulated in vitro with anti-CD3 and CD28 mAbs (5 μg/ml) for 3 days. Representative flow cytometry histograms of CFSE-labeled CD4⁺ T cells are shown. **(J)** Division index was calculated with FlowJo software. Division index of 5 independent experiments is compiled. (*$P < 0.05$ by Student $t$ test). **(K)** Naive CD4⁺ T cells were stimulated with anti-CD3 and CD28 mAbs (5 μg/ml) for 3 days to allow multiple rounds of cell divisions. After day 3, cells were stained with Annexin V-FITC antibody. The percentage of apoptotic cells (Annexin-V⁺) were analyzed by flow cytometry. **(L)** Bar graph shows quantitation of the percentage of apoptotic (Annexin V positive) cells of 5 independent experiments ($n = 5$). (**$P < 0.01$, by Student $t$ test). For numerical raw data, please see S5 Data. BM, bone marrow; CFSE, carboxyfluorescein succinimidyl ester; Cul4b, Cullin-4b; mAbs, monoclonal antibodies; SEM, standard error of mean.

absence of Cul4b, fewer cells entered into the G2-M phase (S5B and S5C Fig), suggesting that S phase progression was defective in Cul4b-deficient T cells.

We then analyzed whether activated Cul4b-deficient CD4⁺ T cells were more sensitive to etoposide, a DNA damage-inducing agent. We found that activated Cul4b-deficient CD4⁺ T cells that were treated with etoposide displayed significantly higher amounts of DNA damage when assessed using a comet assay (Fig 6B and 6C). We further analyzed the activated and etoposide-treated (1 h) CD4⁺ T cells and found that after 40 h of TCR stimulation, Cul4b-deficient CD4⁺ T cells had higher levels of phospho-ser139 H2AX (γH2AX), a well-characterized indicator of a DNA damage (Fig 6D and 6E). Furthermore, when we assessed DNA damage response signaling, we found that following activation, Cul4b-deficient CD4⁺ T cells had elevated levels of p53, p-ATM, and p-CHK1 and which increased further upon etoposide treatment (Fig 6F–6K). These results suggested that Cul4b promotes the DNA damage response in activated CD4⁺ T cells at a time point that correlates with DNA synthesis and proliferation. These data support that, once activated, Cul4b-deficient CD4⁺ T cells were less likely to proliferate and progress through cell cycle and were more prone to apoptosis, likely because of increased DNA damage.

## Cul4b uses DCAF1 as a substrate recognition receptor and interacts with proteins associated with the DNA damage response

The finding that Cul4b was promoting T cell proliferation, survival, and genome stability prompted us to investigate the mechanism(s) underlying its function. Cul4b is known to act as a scaffold protein in a multisubunit protein complex that uses DDB1 as an adaptor protein (Fig 7A). DDB1 has been shown to recruit substrate receptors to Cul4-containing CRL4 complexes [44]. We postulated that Cul4b substrate receptors that are part of an active CRL4B complex would be more likely to get ubiquitinated and hence might accumulate in cells lacking Cul4b. Thus, to begin to identify the substrate receptor/s used by Cul4b in CD4⁺ T cells, we first used mass spectrometry to compare the relative abundances of substrate receptors in control and Cul4b-deficient CD4⁺ T cells. We identified DDB2 as well as 10 DDB1 and CUL4-associated factors (DCAFs) (S6A Fig, S1 Table). Among these putative substrate receptors, Vprbp (DCAF1), Ambra1 (DCAF3), DCAF5, DDB2, DCAF10, and DCAF15 were significantly increased in abundance in Cul4b-deleted CD4⁺ T cells. Four of these, VPRBP

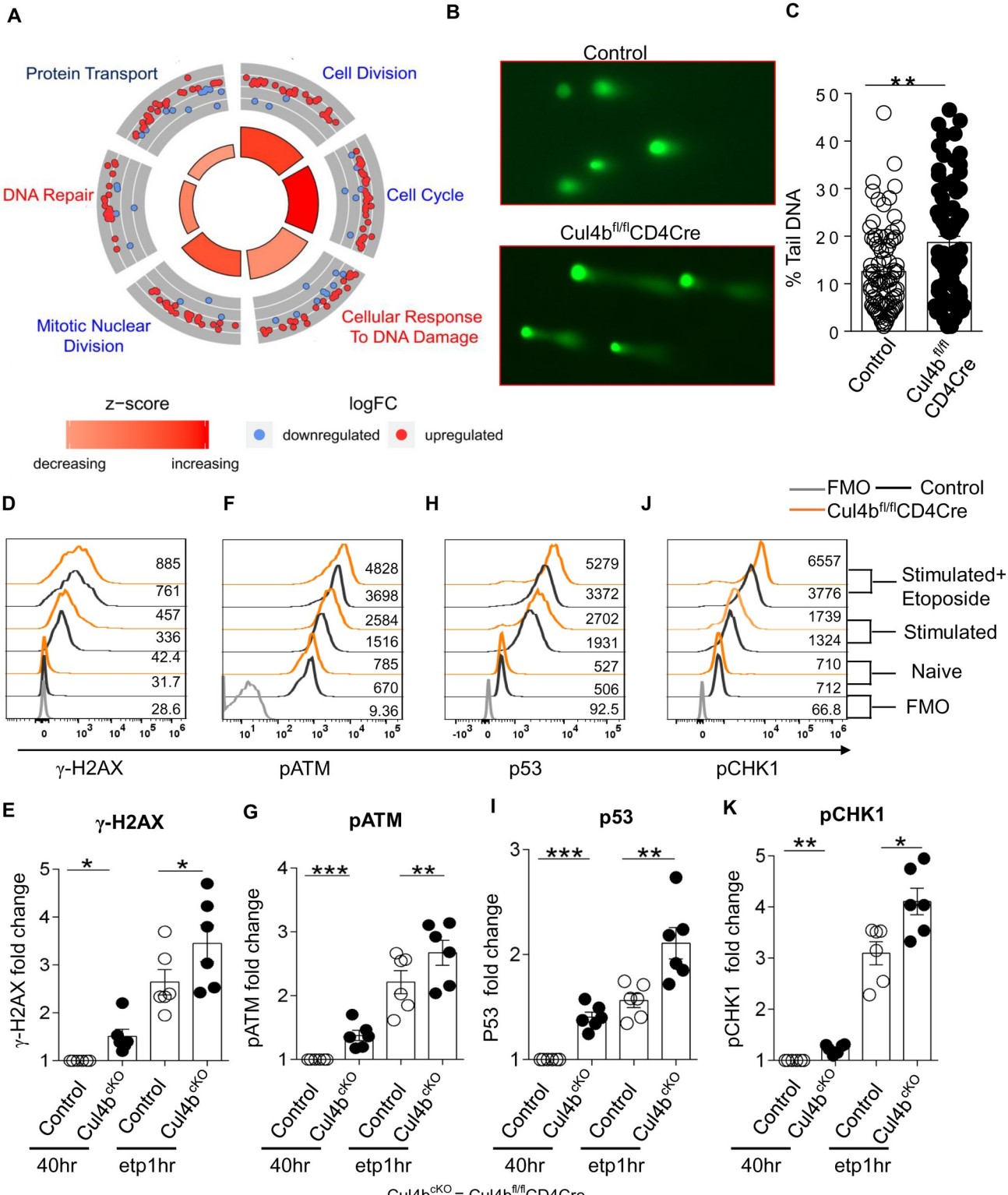

**Fig 6. Cul4b regulates DNA damage response of activated CD4⁺ T cells.** (**A**) Cul4b-deleted (Cul4b^fl/fl^-CD4Cre) and control (Cul4b^fl/fl^) CD4⁺ T cells were cultured for 3 days and then rested in IL-2 for 2 days. After resting, these cells were restimulated for 4 h with anti-CD3/CD28 mAbs (5 μg/ml). Proteins were quantified by iBAQ intensities values and were compared between control and Cul4b-deleted CD4⁺ T cells to generate fold changes. Ontological analysis of the 517 proteins, which showed significant changes in relative abundance between Cul4b^fl/fl^-CD4Cre and control (Cul4b^fl/fl^) CD4⁺ T cells, is represented. The circular plot depicts the enriched functional networks of proteins. The outer circle is a scatter plot for each biological term of

the logFC of the enriched proteins. Within each network, a protein is depicted as a dot, red for up-regulated and blue for down-regulated. The size of the inner trapezoids corresponds to the adjusted *P* value, and the color indicates the z-score as calculated by GOplot algorithm. (**B**) Detection of DNA damage in ETP-treated activated T cells via comet assay. CD4$^+$ T cells were stimulated with anti-CD3/CD28 mAb; on day 2, cells were treated with ETP (10 μM) for 1 h. (**C**) The percent tail DNA was calculated using OpenComet software ($^{**}P < 0.01$, by Student *t* test). (**D–K**) The expression of γ-H2AX, pATM, P53, and pCHK1 was analyzed by flow cytometry. The CD4$^+$ T cells were stimulated with anti-CD3/CD28 mAb for 40 h; after 40 h, cells were either treated with ETP (10 μM) for 1 h or left as such. The expression in unstimulated control and Cul4b$^{fl/fl}$-CD4Cre, 40 h stimulated control and Cul4b$^{fl/fl}$-CD4Cre, and ETP-treated control and Cul4b$^{fl/fl}$-CD4Cre CD4$^+$ T cells is shown. The FMO controls were used to determine positivity. The fold change in fluorescence was calculated by dividing the fluorescence intensity with that of control (40 h stimulated). Bar graph shows the fold change in median fluorescence intensity, *n* = 6. ($^*P < 0.05$, $^{**}P < 0.01$, by Student *t* test). For numerical raw data, please see S6 Data. For supporting dataset, please see S1 Table. Cul4b, Cullin-4b; ETP, etoposide; FMO, fluorescence minus one; iBAQ, intensity-based absolute quantification; IL-2, interleukin 2; mAbs, monoclonal antibodies.

(DCAF1), AMBRA1 (DCAF3), DCAF10, and DCAF15 showed a log2 fold change greater than 1 ($\geq$2-fold) (S6A Fig). To determine the relative abundance of these factors, we assessed their copy numbers and concentration, using the proteomic ruler method detailed above [37]. Among these four, DCAF1 showed the highest number of copies and concentration in control (WT) T cells (S6A Fig, S1 Table), supporting that it may be particularly relevant in T cells. To determine whether this altered protein abundance of substrate receptors was due to decreased transcription, we performed transcriptomic analysis in α-CD3/CD28-stimulated WT and Cul4b-deficient CD4$^+$ T cells. RNA transcripts (FPKM values) of reported substrate receptors were unchanged. This supported that the alteration in protein abundance in Cul4b-deficient T cells was more likely due to increased protein stability (S6B Fig, S2 Table).

DCAF1, when part of an active Cul4b complex, is likely to undergo a higher rate of proteasomal degradation [45]. To test this directly, we analyzed DCAF1 levels in WT and Cul4b-deficient CD4$^+$ T cells that were stimulated with anti-CD3/CD28 for 24 h by WB. We found that both naïve and activated Cul4b-deficient CD4$^+$ T cells had higher levels of DCAF1 (Fig 7B). To assess DCAF1 stability, we first tested whether DCAF1 was degraded in WT CD4$^+$ T cells by assessing protein levels after cycloheximide (CHX) treatment. We found that 4 h of CHX treatment resulted in a substantial loss in the levels of DCAF1 (S6C Fig). We next compared the effects of CHX treatment in WT and Cul4b-deficient CD4$^+$ T cells. We found that DCAF1 was more stable in CD4$^+$ T cells lacking Cul4b (Fig 7C and 7D). Given that Cul4b has been shown to mediate either K48 ubiquitination or monoubiquitination [23,46], we infer that DCAF1 is degraded by the proteasome. Taken together with published data [47], our data support that DCAF1 is very likely to be a key substrate receptor for Cul4b in CD4$^+$ T cells.

To identify other cellular factors that associate with Cul4b and DCAF1 in CD4$^+$ T cells, we analyzed Cul4b and DCAF1-interacting proteins using tandem mass spectrometry (MS/MS). We immunoprecipitated (IP) Cul4b and DCAF1 from CD4$^+$ T cells and analyzed proteins that were IP'ed with Cul4b or DCAF1 but not in the lysates IP'ed with an isotype control. We found that both Cul4b and DCAF1 associated with known components of CRL4 complexes including DDB1 and subunits of the COP9 signalosome (Fig 7E, S3 Table). Of note, we found that DCAF1 was the most enriched Cul4-substrate receptor, based on the number of peptides identified. These data supported that DCAF1 might be a particularly important substrate receptor under these conditions. Additionally, consistent with our previous findings, we found more peptides corresponding to Cul4b than Cul4a associated with DCAF1. Interestingly, we found that both Cul4b and DCAF1 interacted with RAD50 and MRE11a (2 key components of the MRE11–RAD50–NBS1 (MRN) complex, SMC1A and RBBP6 (Fig 7F, S3 Table). The MRN complex has been shown to promote the sensing and repair of DNA damage [48], while RBBP6 is known to regulate replication fork progression and DNA damage at fragile sites [49]. However, given the abundance of pATM and g-H2AX staining in Cul4b-deficient T cells, it was clear that the MRN complex was not defective (Fig 6D–6G). Thus, we sought to further

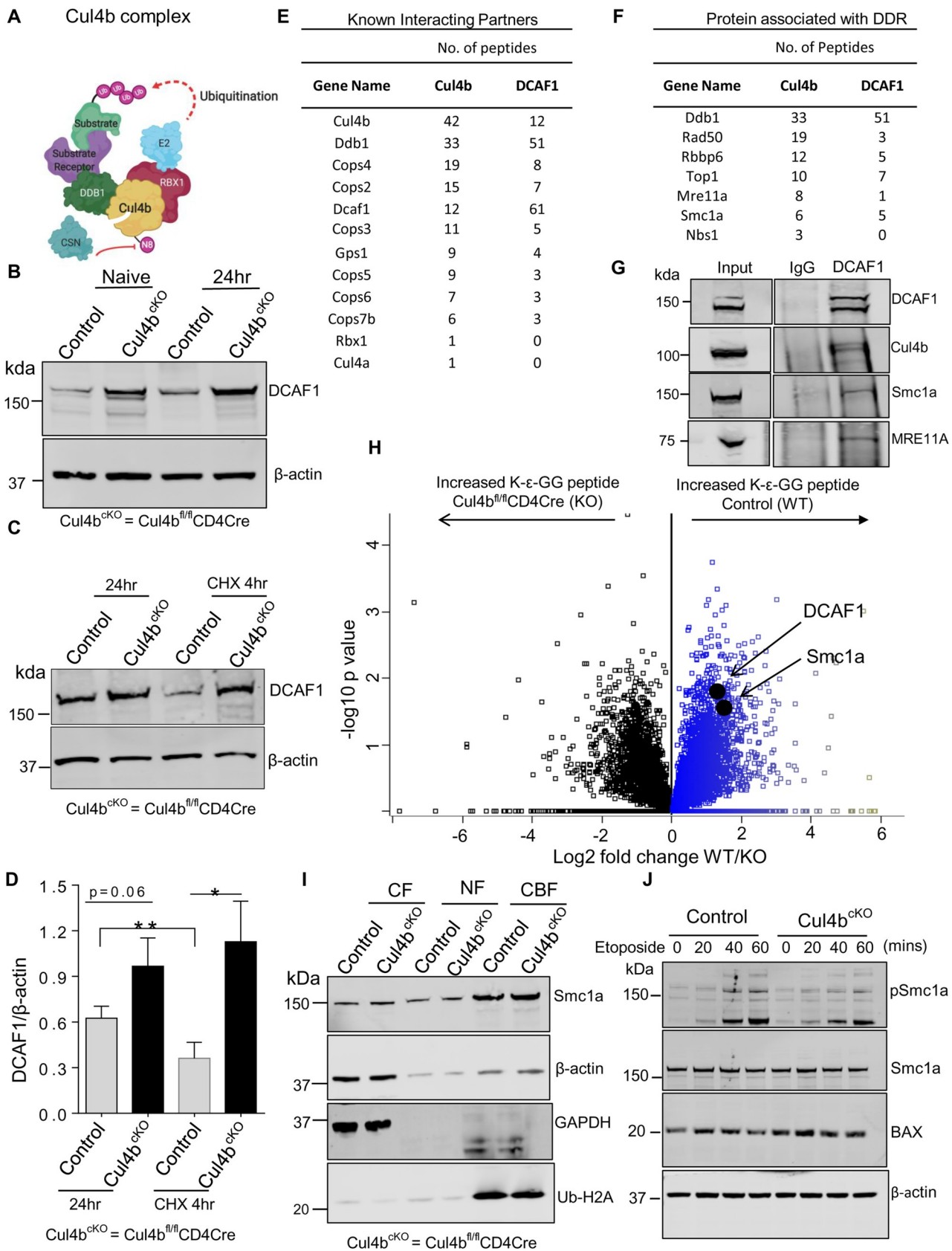

**Fig 7. Cul4b preferentially interacts with DCAF1 to regulate the DNA damage responses in CD4+ T cells.** (**A**) The prototypical Cul4b complex with adaptor protein DDB1, substrate receptor, RING protein, and E2 ligase. (**B**) The expression of DCAF1 was monitored by immunoblotting in naïve and 24 h activated CD4+ T cells; β-actin was used as loading control. (**C**) The stability and protein turnover of DCAF1 in anti-CD3- and anti-CD28-activated (for 24 h) control and Cul4b$^{fl/fl}$-CD4Cre CD4+ T cells in the presence of translation inhibitor CHX was determined by immunoblotting; β-actin was used as loading control. (**D**) The quantitative data of 3 independent experiments are shown; DCAF1 expression in presence and absence of CHX for control and Cul4b$^{fl/fl}$-CD4Cre CD4+ T is shown. (*$P < 0.05$, **$P < 0.01$, by Student $t$ test). (**E**) The endogenous immunoprecipitation coupled with mass spectrometry was done to identify the interacting partners. The proteins preferentially immunoprecipitated by anti-Cul4b and anti-DCAF1 over anti-IgG in TCR-stimulated CD4+ T cells were identified by mass spectrometry. Known interacting partners including DDB1, RBX1, and COP9 signalosome components identified were listed in the table, including the number of unique peptides detected. (**F**) The proteins immunoprecipitated by anti-Cul4b and anti-DCAF1 over anti-IgG and associated with DNA damage response and DNA repair are listed in table (from a single representative). The proteins with more than 5 peptides identified in at least 1 IP were selected except NBS1 which is part of MRN complex (MRE11A-RAD50-NBS1 but was identified only in Cul4b). (**G**) Coimmunoprecipitation of Cul4b, MRE11A, SMC1A with DCAF1 in TCR-stimulated CD4+ T cells. Protein lysates were preincubated with DNase I (10 μg/ml) for 30 min on ice prior to IP. Data are representative of 3 independent experiments. (**H**) Di-glycine remnant profiling was used to identify differentially ubiquitinated peptides in WT and Cul4b-deficient CD4+ T cells restimulated for 4 h with anti-CD3/CD28 mAbs. Di-glycine abundance log$_2$ fold changes for di-glycine peptides in control (WT) and Cul4b$^{fl/fl}$CD4Cre (KO) CD4+ T cells are shown in the volcano plot. (**I**) Control and Cul4b$^{fl/fl}$CD4Cre (Cul4b$^{cKO}$) CD4+ T cells were stimulated for 40 h and then treated with etoposide for 1 h. Cytoplasmic (CF), nuclear-soluble (NF), and chromatin-bound (CBF) proteins were harvested and analyzed by immunoblot. Ubiquitinated histone H2A shows enrichment of CBF, and GAPDH shows enrichment of CF. β-actin shows the amount of protein used for CF and NF. (**J**) Control and Cul4b$^{fl/fl}$CD4Cre CD4+ T cells were stimulated for 40 h and then treated with etoposide for different time points (20, 40, and 60 min). Immunoblot showing expression of pSMC1a, total Smc1a, and BAX. β-actin was used as internal loading control. For numerical raw data, please see S7 Data. For supporting dataset, please see S3 and S4 Tables. CBF, chromatin-bound fraction; CF, cytoplasmic fraction; CHX, cycloheximide; Cul4b, Cullin-4b; DDB1, damaged DNA-binding protein 1; GAPDH, glyceraldehyde 3-phosphate dehydrogenase; IgG, immunoglobulin G; IP, immunoprecipitation; KO, knockout; mAbs, monoclonal antibodies; NF, nucleoplasmic fraction; TCR, T cell receptor; WT, wild-type.

asses SMC1A. SMC1A is a protein that works with the MRN complex and BRCA1 to promote DNA repair and was found to associate with both Cul4b and DCAF1 (Fig 7F). The endogenous interaction of DCAF1 and Cul4b with MRE11 and SMC1a was verified by IP and WB (Fig 7G, S7A Fig). Additionally, when we IP'ed SMC1A, we were able to identify Cul4b and DCAF1 by WB (S7B Fig). It is worth noting that in murine CD4+ T cells, we identified 2 bands for DCAF1 (Fig 7G). Based on our limited analysis of this, the second band becomes more evident 48 h after TCR stimulation. Together, these data support that Cul4b and DCAF1 interact with Smc1a. Notably, the transcript levels of proteins found to be interacting with Cul4b and DCAF1 (*Rad50*, *Nibrin*, *Mre11a*, *Smc1a*, and *Top1*) and reported to be associated with DNA damage response were unchanged (S7C–S7G Fig, S2 Table).

## Cul4b promotes phosphorylation of Smc1a to aid in DNA damage repair

We next sought to identify proteins that were differentially ubiquitinated in control and Cul4b-deficient CD4+ T cells and using di-glycine remnant profiling. In total, 353 peptides showed a significantly increased frequency of di-glycine remnants in Cul4b-sufficient CD4+ T cells. Of these, 198 diglycine-modified (K-ε-GG) peptides were 2-fold higher (log2 >1) in Cul4b-sufficient CD4+ T cells (Fig 7H, S4 Table), supporting that their ubiquitylation may depend either directly or indirectly on Cul4b. We next examined the relationship between proteins with increased diglycine-modified lysine and proteins identified with both Cul4b and DCAF1 in IP-MS/MS to identify the potential substrate(s). We found 5 proteins with an increased diglycine-modified lysine (ubiquitinated) and that were found to interact with Cul4b and DCAF1 (S7H Fig, S4 Table). Among these, we identified DCAF1 and COPS3 proteins which are part of Cul4b complex. Importantly, we also identified SMC1A. To further assess the role of Cul4b-dependent ubiquitination in regulating SMC1A function, we analyzed the ability of SMC1A to get recruited to the chromatin in Cul4b-deficient and control cells. We found that recruitment of SMC1A to chromatin did not depend on Cul4b (Fig 7I). Once SMC1A is recruited to DNA lesions, it is phosphorylated at serine 957 and 966 by ATM, and this has been shown to be crucial for DNA repair and cell survival [50,51]. Thus, we next assessed SMC1A phosphorylation (pSMC1A) using antibody specific for serine 957. We found

that Cul4b-deficient CD4$^+$ T cells had lower levels of pSMC1A. This was particularly noteworthy considering that pATM was significantly higher in these cells compared to control CD4 T cells (Fig 7J). Higher BAX protein levels in Cul4b-deficient cells compared to Cul4b-sufficient cells after activation and etoposide treatment were also observed (Fig 7J).

These data support that, in activated CD4$^+$ T cells, Cul4b and DCAF1 associate with a DNA damage repair complex that includes SMC1A and regulates pSMC1A. Importantly, while SMC1A was identified as differentially ubiquitinated as well as differentially phosphorylated, its protein levels were similar in Cul4b-sufficient and Cul4b-deficient CD4$^+$ T cells. The significance of altered phosphorylation of Smc1a was reflected at the increased expression of proapoptotic genes including *p21*, *Bbc3(puma)*, and *Bax* in Cul4b-deficient T cells (S7I–S7K Fig, S2 Table). Given the role of the MRN complex in lymphocyte survival [52] and the impact of MRN-ATM-dependent phosphorylation of SMC1 in mediating cell survival and chromosomal stability [51], our data support that Cul4b and DCAF1 collaboratively promote the repair of DNA that is damaged during DNA replication, thus promoting the survival and expansion of CD4$^+$ T cells.

## Discussion

Proliferating cells must be equipped with the machinery to sense and rapidly repair DNA damage that occurs during replication. This is particularly true for T cells that must expand at an extraordinary rate to rid the host of an invading pathogen [5,53–56]. Activated T cells have an extremely short cell cycle characterized by correspondingly shortened G$_1$ and S phases, rendering them susceptible to DNA damage. To allow for this, T cells must have a strong and rapid DNA damage response (DDR) to facilitate the repair of damaged DNA and allow cells to expand. While it has been shown that the DDR is provoked by T cell activation [10], little is known regarding how cell cycle progression and DNA damage responses are controlled in rapidly proliferating T cells. In this study, we reveal that activated and proliferating T cells depend on the E3 ubiquitin ligase Cul4b to rapidly repair damaged DNA.

Under homeostatic conditions, mature naïve T cells circulate the body in a quiescent state with a low rate of cell growth and proliferation. Host infection triggers pathogen-specific T cells to exit quiescence and enter cell cycle, followed by rapid proliferation to eliminate the pathogen and to restore homeostasis. We found that levels of both Cul4a and Cul4b were low in naïve T cells but that the activity (neddylation) and the cellular concentration of both Cul4 proteins were significantly increased following TCR stimulation. These data support that both Cul4 proteins are dependent on the same signaling pathways for their induction in T cells. In spite of this, the overall abundance of Cul4a remained much lower than Cul4b. Thus, while Cul4a and Cul4b have been reported to share overlapping functions through targeting the same substrates for ubiquitination [27,46,57], the higher abundance of Cul4b makes it more functionally relevant than Cul4a in T cells. These data likely explain why, in activated CD4$^+$ T cells, the loss of Cul4b cannot be fully compensated for by the presence of Cul4a.

Both Cul4a and Cul4b bind the WD40-like repeat-containing protein DDB1 and use it as an adaptor for recruitment of substrate receptors. DDB1 was initially identified as a damaged DNA-binding protein that heterodimerizes with DDB2 to recognize UV- or mutagen-induced DNA lesions and recruit nucleotide excision repair machinery to remove damaged nucleotides [58]. *Ddb1* deletion in mice resulted in embryonic death at E12.5 [59]. In embryonic fibroblasts, *Ddb1* deletion led to accumulation of DNA damage, rapid induction of apoptosis, and induction of P53 [59,60]. These observations are similar to what we have observed in activated T cells lacking Cul4b. Further supporting the phenotypic similarities between these 2 strains, deletion of *Ddb1* was reported to have no effect on resting mature lymphoid cells but resulted

in apoptosis of cells once they entered the cell cycle [60]. However, that DDB1-deficient T cells were reported to have a more profound loss than we report here with T cells lacking Cul4b supports that Cul4a may promote DNA repair in the absence of Cul4b, albeit at a lower level.

WD40 repeat-containing proteins have the potential to interact with DDB1 and act as substrate receptors within a Cullin RING Ligase 4 (CRL4) complex [44]. Using standard immunoprecipitation technique coupled with mass spectrometry, we identified 11 proteins that interact with Cul4b and that contain a WD40 domain (S3 Table). Among these WD40 repeat-containing proteins, DCAF1, AMBRA1, and DDB2 were significantly increased in Cul4b-deleted CD4$^+$ T cells. Given that proteins within an active CRL complex are likely to have a faster turnover [61], we posited that these 3 proteins are likely substrate receptors that work with Cul4b. Using biochemical and molecular techniques, we validated the interaction between Cul4b and DCAF1. These data support that DCAF1 is a key substrate receptor that cooperates with Cul4b in CD4$^+$ T cells. DCAF1, also known as VprBP (HIV-1 viral protein r-binding protein), is an evolutionary conserved substrate-binding subunit of CRL4 ubiquitin ligases. It is known to be targeted by HIV-1 and 2 viral proteins R and X (Vpr and Vpx), respectively [62,63]. DCAF1 modulates cellular responses against HIV in macrophages [64] and controls cell survival [65–67]. DCAF1 is recruited by RAG1 to ubiquitinate proteins to limit error-prone repair during V(D)J recombination, and development is blocked at the pro-to pre-B cell transition in B cells lacking DCAF1 [68–70]. Loss of DCAF1 has been shown to decrease the numbers of T cells and abolish T cell-mediated antiviral immune response through stabilization of p53 [71]. In Cul4b-deficient T cells, DCAF1 levels were elevated, and yet the cells showed increased apoptosis, supporting that DCAF1 might be working with Cul4b to regulate T cell survival. Interestingly, DCAF1-deficient T cells displayed a more severe defect than what we observed with Cul4b-deleted T cells. This may be because DCAF1 can work with both RING-family CRL4 and HECT-family EDD/UBR5 E3 ubiquitin ligases [72,73]. Nonetheless, our findings together with published data from other labs suggest that destabilizing the CRL4B complex by deleting either adaptor protein (DDB1), substrate receptor (DCAF1) or the E3 ubiquitin ligase (Cul4b), can have a similar impact on T-cell proliferation and survival. Thus, blocking the function of this Cul4b–DDB1-DCAF1 complex could be a useful therapeutic strategy for limiting the numbers of activated T cells in autoimmune diseases. While our data support an important role for DCAF1 as a substrate receptor for Cul4b, we do not rule out the possibility that other substrate receptors may be playing a role in these processes.

The underlying mechanism(s) that link the Cul4b–DDB1-DCAF1 complex to DNA repair remains elusive. Loss of DDB1 is shown to cause accumulation of the cell cycle inhibitor protein Cdkn1a (p21) in bone marrow progenitors and T cells [60]. However, silencing of p21 in *Ddb1*-deleted cells did not rescue hematopoietic stem and progenitor cell (HSPC) numbers, suggesting that DDB1 exerts its function through additional mechanisms. One possibility is that DDB1 targets a substrate that results in, among other things, increased p21 levels. Supporting this, DCAF1 deletion stabilized p53 protein in CD4$^+$ T cells [71]. p21 induction is a well-known outcome of increased p53 activity and is thought to link DNA damage to cell cycle arrest [74]. However, our data support that the CRL4B complex regulates a pathway upstream of p53 that involves DNA damage repair response. Thus, as DNA damage accumulates, p53 becomes activated, resulting in cell cycle arrest and apoptosis. Supporting this, it has been shown that DNA damage and cell cycle processes are interwoven and highly coordinated to maintain genome integrity [75].

Using proteomics and DAVID analysis, we determined that "Cell Cycle" and "DNA damage response" pathways were altered in Cul4b-deficient T cells. Indeed, the cellular processes that were found to be enriched were the same pathways that were altered in Cul4b-deleted

CD4[+] T cells in vitro. We found higher DNA damage in Cul4b-deleted CD4[+] T cells and greater sensitivity to DNA-damaging agents, priming these cells for less proliferation and more apoptosis. We attribute this to the ability of Cul4b and DCAF1 to interact with MRE11A, SMC1A, and other proteins in the DDR pathway. Mre11a is a part of MRN complex (MRE11, RAD50, and NBS1), a sensor of DNA breaks. The Nijmegen breakage syndrome (NBS) is an inherited genetic disorder having mutation in the *NBS1* gene located on chromosome 8q21 [76]. Patients with NBS have reduced T cell numbers [77,78]. Rad50 mutants (Rad50[s/s]) are associated with progressive failure of hematopoietic and germline cells with decreased splenic T and B cells [52]. SMC1A is phosphorylated in an ATM/NBS1-dependent manner [79], and lack of pSMC1A in mammalian cells has been shown to result in defective S-phase checkpoint, decreased survival, and increased chromosomal aberrations after DNA damage [51]. The dependence of pSMC1A on the presence of functional Cul4b places Cul4b downstream of ATM phosphorylation but upstream of SMC1 phosphorylation. Phosphorylated SMC1A is known to colocalize in foci with phospho-ATM and NBS1 [51], supporting that SMC1A might be phosphorylated by ATM at the site of DNA breaks. However, it remains possible that pSMC1A depends on Cul4b marking the DNA damage sites by ubiquitinating histones and associated DDR proteins, thereby regulating their function at the sites of DNA damage. Nonproteolytic ubiquitination has an important regulatory role in DSB signaling and repair; in particular, K63-linked chains are instrumental in recruiting proteins to DSB sites [80]. It is possible that Cul4b is an immediate DSB-responsive ubiquitin ligase that promotes the accumulation or interaction of ATM, SMC1a, and MRN complex at DSB sites via ubiquitination. Supporting this, reduced ubiquitination of histones in Cul4 knockdown cells has been shown to result in a decrease in the activation of DNA repair pathways [46,81]. Another substrate receptor of Cul4-Ddb1 is CDT2 which targets DNA replication factor CDT1. In vertebrates, origin refiring in S, G2, and M phase is prevented primarily through the inhibition of CDT1 activity via Geminin and ubiquitin-mediated proteolysis by Cul4 [82]. Therefore, given that Cul4b can be important in DNA replication licensing and maintaining the structural integrity of prereplication complexes [83], it is possible that Cul4b regulates multiple pathways linking replication, DNA damage, and DNA damage repair. Our data support that the association of Cul4b with the MRN complex is critical for allowing SMC1A to resolve the DNA damage and promote cell survival. Taken together, our data support that Cul4b is critical for allowing CD4[+] T lymphocytes to sustain their unique bursts of proliferation after activation by resolving DNA damage.

## Methods

### Ethics statement

All procedures were conducted in accordance with the Animal Welfare Act and were approved by the Institutional Animal Care and Use Committee at the Children's Hospital of Philadelphia (protocol ID 810).

### Mice

The CRISPR/Cas9 system was used to insert the loxP sites between the exons 3 and 4 of Cul4b gene. The F1 generation of Cul4bfl/fl mice were produced by a cross between a C57BL/6J (B6) x an SJL/J (SJL) mice. The mice were backcrossed for 10 generations to generate the C57BL/6J (B6) inbred strain. Rag1−/−, CD4Cre, and CD45.1 congenic wild-type mice were on C57BL/6 background. All mice were bred in house under specific pathogen-free conditions in the animal facility at the Children's Hospital of Philadelphia (CHOP). Mice between 8 and 14 wk of age were used, and within experiments they were age, gender, and cage matched. For

simplicity, Cul4b[fl/fl] CD4-Cre refers to both male (Cul4b[fl/Y] CD4-Cre) and female mice that lack Cul4b in T cells. Animal housing, care, and experimental procedures were performed in compliance with the CHOP Institutional Animal Care and Use Committee.

## Flow cytometry

Thymus, lymph nodes, and spleen were mechanically dissociated and macerated through the 70-μm cell strainer. For colon, single-cell suspension was prepared using enzymatic digestion (Collagenase/DNase) at room temperature for 1 h. Single-cell suspensions were stained with a fixable viability dye, then pretreated with unlabeled anti-CD16/CD32 (Fc Block BD Pharmingen). Cells were then stained in FACS buffer (phosphate-buffered saline (PBS) containing 2.5% fetal calf serum and 0.1% sodium azide) with mixtures of directly conjugated antibodies (CD3 (Clone 17A2, BD Bioscience), CD4 (Clone GK1.5 and RM4-5 Biolegend) CD8, (Clone 53–6.7 Biolegend) B220 (Clone RA3-6B2 BD Bioscience), CD44 (Clone 1M7 Biolegend), CD62L (Clone MEL-14 Biolegend), CD69 (Clone H1.2F3 BD Bioscience), and CD25 (Clone PC61.5 eBioscience).

For expression of intracellular antigens [Ki67 (Clone 16A8 Biolegend), γ-H2AX (Clone 2F3 Biolegend), pATM (Clone 10H11.E12 Biolegend), pCHK1 (Clone 133D3 Cell Signaling Technology), and P53 (Clone 1C12 Cell Signaling Technology)] Foxp3/Transcription Factor Staining Buffer (Thermo Fisher Scientific) Set was used. The staining procedure was carried out as per manufacturer's instruction with some modifications; cells were stained for 2 h at 4˚C. The Fluorescence Minus One (FMO) controls were used to set gates and determine positive staining.

For the T cell proliferation assay, naïve CD4[+] T cells were isolated from control (CD45.1) and Cul4b[fl/fl]CD4-Cre (CD45.2) and were cocultured for 3 or 5 days. Briefly, CD4[+] T cells (100,000/well) were incubated with CFSE (Thermo Fisher Scientific) at a final concentration of 5 μM for 10 min at 37˚C. Cells were washed 3 times with ice-cold complete RPMI 1640 and stimulated with anti-CD3 (Clone 17A2 Biolegend)/CD28 (Clone 37.51 Biolegend) antibodies (5 μg/ml). After day 3, cells were harvested and stained with anti-CD45.1 (Clone A20 Biolegend) and anti-CD45.2 (Clone 104 Biolegend) antibodies, and change in CFSE intensity was measured, while the relative percentages of cells in coculture were analyzed at day 5. Samples were analyzed using a Fortessa (BD Biosciences) flow cytometer at the CHOP flow cytometry core facility. Data were analyzed using FlowJo software V10 (TreeStar Ashland). Results are expressed as the percentage of positive cells or median fluorescence intensity (MFI). All FCS files are available on flowrepository platform (https://flowrepository.org/) using ID: FR-FCM-Z38M.

## Bone marrow chimeras

Bone marrow (BM) chimeras were generated by irradiating Rag1[−/−] recipient mice using an X-Rad Irradiator; sublethal irradiation of 400 rad was used. The next day, BM from donor mice was collected, RBCs were lysed, and T cell depletion was done using Phycoerythrin (PE)-conjugated TCR-β antibody (Clone H57-597 Biolegend) and anti-PE microbeads (Miltenyi Biotec). Briefly, BM cells were isolated and resuspended in MACS buffer stained with PE-conjugated TCR-β antibody and anti-PE microbeads. After staining, cell suspension was loaded onto the column to deplete BM cells of T cells; flow through was collected and used for the experiments. T cell-depleted BM cells from control (CD45.1) and Cul4b[fl/fl]-CD4Cre (CD45.2) were enumerated and mixed at a 1:1 ratio. A total of 2 million mixed BM cells in plain media were injected into recipient mice via tail vein injections. The irradiated mice were kept on Sulfatrim antibiotic water for 3 wk. Mice were analyzed ≥8 wk after injection of BM cells.

## CD4<sup>+</sup> T cell isolation and in vitro stimulation

Naïve CD4[+] T cells were isolated by magnetic separation using the Miltenyi naïve CD4[+] T cell isolation kit. Briefly, cells were isolated from spleen and lymph nodes. Single-cell suspension was resuspended in MACS buffer and stained with antibody-conjugated beads. For naïve CD4[+] T cells isolation, a cocktail of biotin-conjugated monoclonal antibodies (mAbs) against CD8a, CD11b, CD11c, CD19, CD25, CD45R (B220), CD49b (DX5), CD105, Anti-MHC class II, Ter-119, and TCRγ/δ was used. Then, microbeads conjugated to monoclonal anti-biotin antibodies and CD44 microbeads were added. After staining, cell suspension was loaded onto the column, and cells that flow-through the column (unlabeled cells) were the enriched naïve CD4[+] T cells.

Naïve CD4[+] T cells were stimulated in vitro in complete RPMI (RPMI 1640 supplemented with 10% fetal bovine serum (Atlanta Biologicals), HEPES (Thermo Fisher Scientific), nonessential amino acids, sodium pyruvate (Thermo Fisher Scientific), 2mM-Glutamine, antibiotics, and 2-mercaptoethanol) with plate-bound anti-CD3 (Clone 17A2 Biolegend) and anti-CD28 (Clone 37.51 Biolegend) antibodies. The tissue culture plates were coated with the antibodies overnight at 4˚C (5 μg/ml). Cells were cultured at 37˚C with 10% $CO_2$.

## Apoptosis and cell cycle analysis

The annexin staining was performed to determine ability of cells to survive under in vitro conditions. In brief, naïve CD4[+] T cells were stimulated with anti-CD3/CD28 mAb (5 μg/ml) for 72 h to generate asynchronous population. After 72 h, cells were harvested, suspended in annexin binding buffer (10 mM HEPES (pH 7.4), 150 mM NaCl, 0.25 mM $CaCl_2$), and incubated with FITC-conjugated Annexin V (Thermo Fisher Scientific) in the dark for 15 min at room temperature. After incubation, 400 μl annexin binding buffer was added and cells were analyzed.

For cell cycle analysis, CD4[+] T cells were incubated in the 24-well culture plate with α-CD3/CD28 mAb (5 μg/ml) for 72 h to generate asynchronous population. After 72 h, cells were harvested and fixed by adding chilled 70% ethanol dropwise under constant vortexing. Cells were washed and stained with FxCycle PI/RNase Staining Solution (Thermo Fisher Scientific) as per manual instructions. Cells were incubated at room temperature for 30 min, and cell cycle was assessed on Fortessa (BD Biosciences) flow cytometer and analyzed using FlowJo software V10 (TreeStar Ashland). All FCS files are available on flowrepository platform (https://flowrepository.org/) using ID: FR-FCM-Z38M.

## Comet assay

Neutral comet assay (Abcam) was used to detect DSBs and DNA lesions as per manufacturer's instructions with slight modifications. Briefly, CD4[+] T cells were stimulated with anti-CD3/CD28 mAb for 40 h, and for the last 1 h in culture, etoposide (Sigma-Aldrich) (ETP; 20 μM) was added to generate detectible amount of DNA damage. Cells were harvested and mixed with low melting temperature agarose at 1/10 ratio (v/v) and layered on slides precoated with 1 layer of agarose. Slides were lysed in lysis buffer (containing NaCl, EDTA, 1% DMSO) overnight at 4˚C and then electrophoresis in prechilled TBE electrophoresis buffer at 1 Volt/cm for 15 min. Slides were washed with prechilled DI $H_2O$ and 70% ethanol and then air dried. The slides were stained with Vista Green DNA Dye for 15 min and visualized under Evos FL Auto epifluorescence microscopy (Thermo Fisher Scientific). Analysis was performed with Open-Comet software v1.3.1. Tail DNA % was measured to show the extent of DNA damage. At least 30 cells per sample were analyzed.

## Adoptive T cell transfer

Control or Cul4b[fl/fl]CD4-Cre mice were killed; lymph nodes and spleens were harvested. Harvested tissues were mashed through 70-μm cell strainers using cold PBS. Red blood cells were lysed using ACK lysis buffer, and single-cell suspension was collected. Naïve $CD4^+$ T cells ($CD4^+$ $CD62L^+$ $CD25^-$ $CD44^-$) were isolated from these cells. The purity of naïve T cells was >95%. Naïve $CD4^+$ T cells were transferred into $Rag1^{-/-}$ mice ($1 \times 10^6$ cells/mouse) via intraperitoneal injection (IP). Body weights were recorded per week. When loss of body weight exceeded 20% after transfer, the host mice were sacrificed. A total of 1 to 2 mm of colon were subjected to pathological observations, and cells from spleen and rest of colon were collected and analyzed by flow cytometry as described above.

## Histological analysis

Stool from colons was removed by lightly pressing it out using curved forceps. A total of 1 to 2 mm sections of the most distal colon were taken and fixed in 10% neutral buffered formalin. Tissues were then paraffin-embedded and sectioned to 5 μm thickness and stained with hematoxylin and eosin (H&E). Images were obtained using a Leica DM4000B upright scope paired with a Spot RT/SE Slider camera with 10× objective lens.

## Western blotting

Cells were washed with PBS ($Ca^{2+}$ and $Mg^{2+}$-free) and lysed using 1% Triton-X lysis buffer containing 50 mM Tris, 50 mM NaCl, Protease inhibitor cocktail (Roche), Halt phosphatase inhibitor cocktail (Thermo Fisher Scientific), $Zn^{2+}$-chelator *ortho-phenanthroline* (o-PA) (LifeSensors), deubiquitylase inhibitor PR-619 (LifeSensors), and 1% NP 40. Protein lysate was harvested and quantitated using Bradford reagent. For Cul4a and Cul4b neddylation experiments, whole-cell lysates were prepared in SDS sample buffer (62.5 mM Tris-HCl, 2% w/v SDS, 10% glycerol, 50 mM DTT, and 0.1% bromophenol blue), vortexed to reduce sample viscosity, denatured by boiling, and then cooled on ice. This method prevents the deneddylation during protein extraction. Samples were resolved on 4% to 12% Novex Tris-Glycine Gels (Thermo Fisher Scientific) and then transferred onto PVDF membrane (Amersham Pharmacia Biotech, Piscataway, New Jersey). The membrane was probed with the primary antibodies Cul4b (Sigma Aldrich;1:500 and ProteinTech;1:1,000), Cul4a (ProteinTech;1:1,000), DCAF1 (Cell Signaling Technology;1:2,000 and ProteinTech;1:1,000), MRE11A (Novus Biologicals;1:1,000), Smc1a (Bethyl Laboratories;1:2,000), phospho-Smc1a (Abcam;1:1,000), UbH2A (Cell Signaling Technology;1:2,000), GAPDH (Milipore;1:5,000), H3 (Cell Signaling Technology;1:1,000), and β-actin (Santa Cruz;1:5,000) as loading control. Immunostaining was performed using appropriate secondary antibody at a dilution of 1:5,000 and developed with licor odyssey imaging system.

## Subcellular fractionation

Subcellular fractionation was carried out as described by Li and colleagues [85] with some modifications. Anti-CD3/CD28-stimulated (40 h) cells were harvested and washed twice with PBS ($Ca^{2+}$ and $Mg^{2+}$-free) and resuspended in solution A (10 mM HEPES (pH 7.4), 10 mM KCl, 1.5 mM $MgCl_2$, 0.34 M sucrose, 10% glycerol, 1 mM dithiothreitol, 10 mM NaF, 1 mM $Na_2VO_3$, EDTA-free complete protease inhibitor cocktail (Roche), deubiquitylase inhibitors PR-619 (LifeSensors), and $Zn^{2+}$-chelator o-PA (LifeSensors). Triton X-100 was added to a final concentration of 0.1%, and the cells were incubated for 5 min on ice. Cytosolic proteins (CF) were separated from nuclei by centrifugation (4 min, $1,500 \times g$). Nuclei were washed

once in solution A, and then lysed in solution B (3 mM EDTA, 0.2 mM EGTA, 1 mM dithio-threitol, protease inhibitors, deubiquitylase inhibitors PR619, and $Zn^{2+}$-chelator o-PA) for 30 min. Insoluble chromatin was then separated from soluble nuclear proteins (NF) by centrifugation (4 min, $1,700 \times g$), washed once in solution B, and collected by centrifugation (1 min, $10,000 \times g$). The final chromatin pellet was resuspended in Triton-X buffer containing DNase (10 µg/ml) to release chromatin-bound proteins. After incubation at 4˚C for 30 min, the digestion was stopped by the addition of 5 mM EDTA, and proteins were resuspended in SDS sample buffer.

## Cycloheximide chase assay

Cycloheximide chase analysis was carried out to check the protein stability of DCAF1. Briefly, $CD4^+$ T cells were isolated and resuspended in complete RPMI (cRPMI) at a concentration of $1 \times 10^6$ cells per ml. Cells were seeded on 24-well plates coated with anti-CD3/CD28 mAbs overnight at 4˚C. After 20 h, stimulation cells were either supplemented with CHX (50 µg/ml, Sigma Aldrich C4859) for different time points or left untreated. CHX was added to wells to inhibit translational. After CHX treatment, cells were washed and cell pellets were lysed in 1% Triton-X lysis buffer and immunoblotting was carried out as described above.

## Immunoprecipitation

Cells were harvested, washed in PBS, and resuspended in Triton-X lysis buffer (1% Triton-X, 50 mM Tris (pH 8), 100 mM NaCl, EDTA-free complete protease inhibitor cocktail (Roche), deubiquitylase inhibitors PR619 (LifeSensors), $Zn^{2+}$-chelator o-PA (LifeSensors), Phosphatase Inhibitor Cocktail). Total protein concentration was determined by Bradford assay. Typically, 2 to 3 mg of protein extract was added to prewashed Dynabeads with 5 to 8 µg of antibody. After overnight incubation, the beads were washed 4 times in PBS-Tween (PBST, 0.2% Tween), and to elute protein, 2x SDS loading buffer was used. Cell lysates were preincubated with DNase I (10 µg/ml Millipore) to digest any DNA prior to immunoprecipitation, to rule out that the interaction is not bridged by DNA.

## Real-time PCR

Total RNA was extracted from sorted naïve $CD4^+$ T cells and stimulated $CD4^+$ T cells using TRIzol reagent (Thermo Fisher Scientific). cDNA was synthesized using the High Capacity RNA-to-cDNA kit according to the manufacturer's protocol. Real-time PCR was performed using TaqMan Gene Expression Master Mix (Thermo Fisher Scientific). Amplification and detection were performed on ABI 7500 Prism Detection System (Applied Biosystems). The mRNA was quantified using the $2^{-\Delta\Delta Ct}$ method, in which Ct represents the threshold cycle. Relative gene expression was determined by normalizing the gene expression of Cul4b to β-actin. The following primers were used to amplify Cul4b (Mm00518513_m1; Thermo Fisher Scientific) and β-actin (Mm00607939_s1; Thermo Fisher Scientific).

## Sample preparation for whole cell proteome

$CD4^+$ T cells were isolated by MACS from control and Cul4bfl/fl-CD4Cre mice. Cells were stimulated for 3 days and rested in IL-2 (50 IU/ml) for 2 days. After rest, cells were stimulated with anti-CD3/CD28 mAbs for 4 h, and cell pellets were stored at −80˚C. For whole cell protein analysis, sample preparation was done as described by Mertins and colleagues [86]. Briefly, cell pellets were lysed using urea buffer (8 M urea, 75 mM NaCl, 50 mM Tris HCl (pH 8.0), 1 mM EDTA, 21 µM aprotinin, 307 nM leupeptin, 1 mM PMSF, 10 mM NaF, 5 mM

sodium butyrate, 5 mM iodoacetamide, and 25 μM PR-619). Protein concentration was determined by a MicroBCA assay. The protein samples were reduced with 5 mM dithiothreitol for 45 min followed by alkylation with 20 mM iodoacetamide for 45 min in the dark. Proteins from each sample were then diluted with 50 mM Tris HCl (pH 8.0) to reduce urea concentration to 1.1 M before trypsin digestion. Trypsin digestion was done overnight at 37˚C. The peptides were desalted with the SepPak columns conditioned with acetonitrile and equilibrated with trifluoroacetic acid. After loading, tryptic peptides were desalted with 0.1% trifluoroacetic acid. Peptides were eluted from the column with 80% acetonitrile/0.1% trifluoroacetic acid. Eluted peptides were lyophilized and stored at −80˚C.

Peptides were fractionated by basic Reverse Phase-High Performance Liquid Chromatographic (RP-HPLC); the peptides prepared were reconstituted in 20 mM ammonium formate (pH 10.0). They were centrifuged at 20,000 $g$ for 5 min before being transferred into an auto sampler. Peptides concentration was determined with a NanoDrop under UV280 before they were injected to HPLC instrument. RP-HPLC was conducted on a Zorbax 300 Å Extend-C18 4.6 mm × 250 mm column (Agilent, 3.5 μm bead size). Prior to peptides separation, the column was monitored for efficient separation with standard mixtures of BSA and Fetuin in house tryptic digests. Solvent A (2% acetonitrile, 5 mM ammonium formate (pH 10)) and solvent B (90% acetonitrile, 5 mM ammonium formate (pH 10)) were used to separate peptides based on their hydrophobicity. For separation of approximately 3.8 mg total peptides, a flow rate of 1 ml/min was used, and the increased percentage of solvent B in a nonlinear gradient is the same as published before [86]. Peptide samples were combined into 6 fractions to be used for proteome analysis. Subfractions were collected in a serpentine, concatenated pattern to generate subfractions of similar complexities that contain both hydrophilic as well as hydrophobic peptides. A total of 5% of the volumetric sample was taken from each subfraction for proteome analysis. These peptide subfractions were subsequently dried by lyophilizition. For proteomics analysis, the peptides were dissolved in water/0.1% trifluoroacetic acid (TFA), and 2 μg of peptides were injected to the liquid chromatography–mass spectrometry (LC-MS).

For diglycine remnant profiling, K-ε-GG peptides were enriched using antibody specific for diglycine peptide; the peptides from the 6 fractions (95% of the total peptides) were recombined noncontiguously into 3 fractions with approximately 1 mg peptide per fraction for immunoprecipitation. PTMscan ubiquitin remnant antibody, noncovalently conjugated to beads (Cell Signaling Technologies), was cross-linked as described [87] and validated by SDS-PAGE. Cross-linked antibody (approximately 31 μg) was used for each 1 mg peptide fraction. The beads and the peptides were undermixed end to end in IAP buffer (50 mM MOPS, 50 mM NaCl, 10 mM Na2HPO4 (pH ~7)) for 1 h at 4˚C. The beads were washed twice with 1 mL IAP, then with 0.5 mL of 0.05% RapiGest SF surfactant in IAP, followed with final washes (3 times) of 1 mL PBS; the lysine acetylated peptides were eluted with 40 μL 0.15% TFA. The eluted peptides were desalted via Oasis HLB uElution plate 30 μM. The eluted samples were dried overnight in speed vacuum dyer, and they were prepared in 0.1% TFA/water for LCMS analysis. HRM peptides (Biognosys) were all included in the samples for LC-MS/MS.

## Mass spectrometry (MS) data acquisition

Tryptic digests were analyzed by LC-MS/MS on a QExactive HF mass spectrometer (Thermo Fisher Scientific, San Jose, California) coupled with an Ultimate 3000. For proteome analysis, peptides were separated by RP-HPLC on a nanocapillary column, 75 μm id × 25 cm 2um PepMap Acclaim column. Mobile phase A consisted of 0.1% formic acid (Thermo Fisher Scientific) and mobile phase B of 0.1% formic acid/acetonitrile. Peptides were eluted into the mass spectrometer at 300 nL/min with each RP-LC run comprising a 90-min gradient from 10% to

25% B in 65 min, 25% to 40% B in 25 min. The mass spectrometer was set to repetitively scan m/z from 300 to 1,400 (R = 240,000) followed by data-dependent MS/MS scans on the 20 most abundant ions, minimum AGC 1e5, dynamic exclusion with a repeat count of 1, repeat duration of 30 s, and normalized collision energy (NCE) of 30 (R = 15,000). FTMS full scan AGC target value was 3e6, while MSn AGC was 1e5, respectively. MSn injection time was 160 ms; microscans were set at 1. Rejection of unassigned and 1, 7, 8, and >8 charge states was set.

## Immunoprecipitation and solution digest for immunoprecipitated samples

To identify Cul4b and DCAF1-interacting proteins in T cells, CD4$^+$ T cells were purified by MACS and activated with anti-CD3/CD28 mAbs for 40 h. Cells were lysed in Triton-X lysis buffer (1% Triton-X, 50 mM Tris (pH 8), 100 mM NaCl, containing EDTA-free complete protease inhibitor cocktail (Roche), deubiquitylase inhibitors PR619 (LifeSensors), and Zn2+-chelator o-PA). The lysates were cleared by centrifugation at 15,000 rpm at 4˚C for 12 min. The supernatant was divided into 3 parts and incubated with Dynabeads (Invitrogen) conjugated either with Cul4b antibody or DCAF1 antibody or rabbit IgG in cold room overnight. The immunocomplex was washed 4 times with PBS-Tween (PBST, 0.2% Tween), and each sample was eluted from beads with 0.3% SDS and acetone/TCA precipitated. The resulting pellet was digested with the iST kit (PreOmics GmbH, Martinsried, Germany) per manufacturer's protocol [88]. Briefly, solubilization, reduction, and alkylation was performed in sodium deoxycholate (SDC) buffer containing TCEP and 2-chloroacetamide. Samples were enzymatically hydrolyzed for 1.5 h at 37˚C by LysC and trypsin. To stop the digestion, the reaction mixture was acidified with 1% TFA in isopropanol. Peptides were desalted and eluted and dried by vacuum centrifugation, then reconstituted in 0.1% TFA containing iRT peptides (Biognosys).

## Immunoprecipitation MS data collection

Peptides from the IP digest were separated by RP-HPLC on Easy-Spray RSLC C18 2 um 75 μm id × 50 cm column at 50C. Mobile phase A consisted of 0.1% formic acid and mobile phase B of 0.1% formic acid/acetonitrile. Peptides were eluted into the mass spectrometer at 300 nL/min with each RP-LC run comprising a 90-min gradient from 1% to 5% B in 15 min, 5% to 45% B in 90 min. The mass spectrometer was set to repetitively scan m/z from 300 to 1,800 (R = 120,000), followed by data-dependent MS/MS scans (R = 45,000) on the 20 most abundant ions, NCE of 27, and dynamic exclusion of 15 s with a repeat count of 1. FTMS full scan AGC target value was 5e5, while MSn AGC was 1e5, respectively. MS and MSn injection time was 120 ms; microscans were set at 1. Unassigned, 1, 6 to 8, and >8 charge states were excluded from fragmentation

## MS raw files search

Protein and peptide identification/quantification was performed with MaxQuant (1.6.3.4) [89] using a mouse reference database from Uniprot (reviewed canonical and isoforms; downloaded on 20180104) appended with the sequence of HRM peptides. Carbamidomethyl of Cys was defined as a fixed modification. Oxidation of Met and acetylation of protein N-terminal were set as variable modifications. Trypsin/P was selected as the digestion enzyme, and a maximum of 3 labeled amino acids and 2 missed cleavages per peptide were allowed. The false discovery rate for peptides and proteins were set at 1%. Fragment ion tolerance was set to 0.5 Da. The MS/MS tolerance was set at 20 ppm. The minimum peptide length was set at 7 amino acids. The rest of the parameters were kept as default. The quality of the generated results by MaxQuant was further verified by Proteomics Quality Control (PTXQC) [90].

## Bioinformatics analysis

Perseus (1.6.2.3) was used for statistical and bioinformatics analysis [91]. First, we removed the contaminants and decoys information from the data set. Subsequently, the protein intensity values were $log_2$ transformed and normalized to the median for each sample. After assigning the categorical annotation, data were filtered based on 3 values in at least 1 group. To quantify the changes in WCP protein expression, iBAQ intensities were used as described earlier by Dybas and colleagues [30]. Student $t$ test was employed to identify up/down-regulated proteins, and lists of differentially expressed proteins with $P$ value < 0.05 were selected and used for further bioinformatics analysis. The gene ontology annotation and functional analysis were performed using DAVID bioinformatics resources (v.6.8). Copy numbers and concentration of proteins were calculated using proteomic ruler method which uses the mass spectrometry signal of histones as an internal standard [37]. GOplot algorithm was employed to visualize the gene ontology analysis results [92]. The mass spectrometry proteomics data have been deposited to the ProteomeExchange Consortium via the PRIDE [93] partner repository with the data set identifier PXD017699 and PXD019272 [94].

## RNA-seq

Naïve CD4+ T cells were isolated by MACS from control and Cul4bfl/fl-CD4Cre mice. Cells were stimulated for 24 h with anti-CD3/CD28 mAbs. Total RNA was isolated using TRIzol reagent (Thermo Fisher Scientific), and poly-A selection was used to remove ribosomal RNA. After first and second strand synthesis from a template of poly-A selected/fragmented RNA, other procedures from end-repair to PCR amplification were performed. The DNA library was quantitated using Qubit. Libraries were sequenced on BGIseq500 platform with 50 bp single-end sequencing.

Single-end sequencing reads were quality checked using FASTQC (www.bioinformatics. babraham.ac.uk/projects/fastqc), and reads were aligned to the Ensembl Mus musculus reference genome (GRCm38) using the spliced transcripts alignment to a reference (STAR) alignment program [95]. The transcripts were assembled using cufflinks, and FPKM table was computed using cuffnorm [96].

## Statistical analysis

Results were expressed as the mean ± standard error of mean (SEM). Statistical analysis was performed using Prism software 6, and $P$ value was calculated using the Student $t$ test. Two-sided $P$ values < 0.05 were considered statistically significant.

## Supporting information

**S1 Fig. TCR-driven activation of Cul4a and Cul4b.** Naive CD4+ T and CD8+ T cells from control mice (C57BL/6 mice) were stimulated with anti-CD3 and anti-CD28 mAbs. Expression of Cul4a and Cul4b was monitored by immunoblotting. (**A–C**) The quantitative data of 4 independent experiments are shown. (**B**) shows inactive (nonneddylated) Cul4b in CD4+ and CD8+ T cells at different time points. (**C**) shows inactive Cul4a in CD4+ and CD8+ T cells. Data were quantitated using Image J software and are represented as mean ± SEM (*$P$ < 0.05 **$P$ < 0.01, ***$P$ < 0.001 by Student $t$ test; ns, not significant, $P$ > 0.05 by Student $t$ test). (D) The activated CD4+ T cells were lysed using different conditions as described above, and presence of neddylated and nonneddylated forms of the protein were determined by immunoblotting. For numerical raw data, please see S1 Data. Cul4a, Cullin-4a; Cul4b, Cullin-4b; mAbs,

monoclonal antibodies; SEM, standard error of mean.
(TIF)

**S2 Fig. Cul4b is dynamically regulated in CD4$^+$ T cells. (A)** Naive CD4$^+$ T cells from control mice were activated either with anti-CD3 mAb or anti-CD3 and anti-CD28 mAbs. In case of anti-CD3/CD28 mAb stimulation, cells were either neutralized for IL-2R by adding anti-IL-2R antibody (10 μg/ml) or left as such. The mRNA expression of Cul4b was detected by RT-PCR. The Cul4b expression was determined in naive and activated CD4$^+$ T cells, and β-actin was used as internal control. Data are represented as mean ± SEM of 3 independent experiments. (**B**) Quantification of protein abundance of Cul4a and Cul4b in CD4$^+$ T cells using mass spectrometry. The copy numbers of Cul4a and Cul4b in naïve and TCR-activated (24 and 48 h) CD4$^+$ T cells are shown and were calculated using the proteomic ruler method. (**C**) The bar graph shows the copy numbers of Cul4a and Cul4b in naïve and antigen-stimulated CD4$^+$ T cells; the data were analyzed from the data set reported by Howden and colleagues [38]. For numerical raw data, please see S2 Data. Cul4b, Cullin-4b; IL-2R, IL-2 receptor; mAbs, monoclonal antibodies; RT-PCR, real-time PCR; SEM, standard error of mean; TCR, T cell receptor.
(TIF)

**S3 Fig. T Cul4b deletion is dispensable for mature cells. (A)** The numbers of various T-cell populations in the thymus of control (Cul4b$^{fl/fl}$) and Cul4b$^{fl/fl}$-CD4Cre mice. The bar graphs show the mean ± SEM of 4 sets of mice (ns, not significant, $P > 0.05$ by Student $t$ test). (**B and C**) The percentages of various T-cell populations in the lymph of control (Cul4b$^{fl/fl}$) and Cul4b$^{fl/fl}$-CD4Cre mice were assessed by flow cytometry. The bar graphs show the mean ± SEM of 8 sets of mice (**$^{**}P < 0.01$ by Student $t$ test; ns, not significant, $P > 0.05$ by Student $t$ test). The mice were paired with respective to age, gender, cage, and time of takedown. (**D and E**) The numbers of various T-cell populations in the spleen and lymph nodes of control (Cul4b$^{fl/fl}$) and Cul4b$^{fl/fl}$-CD4Cre mice are shown. The bar graphs show the mean ± SEM of 4 sets of mice (ns, not significant, $P > 0.05$ by Student $t$ test). For numerical raw data, please see S3 Data. Cul4b, Cullin-4b; SEM, standard error of mean.
(TIF)

**S4 Fig. Cul4b regulates homeostasis, proliferation, and survival of activated CD4$^+$ T cells.** (**A**) The comparison of the total CD4$^+$ and B cell populations in the spleen and lymph nodes of irradiated recipient chimeric mice after reconstitution of bone marrow cells from control mice (CD45.1$^+$) and Cul4b$^{fl/fl}$-CD4Cre mice (CD45.2$^+$). (**B and C**) The line graphs show the relative ratios of CD4$^+$ T cells in spleen and lymph nodes. Ratios were calculated by dividing the percentages of CD4$^+$ T cells of each genotype with the percentages of B cells from the same genotype. The pie chart depicts the relative percentages of the control and Cul4b$^{fl/fl}$-CD4Cre CD4$^+$ T cells. (**D–G**) CD4$^+$ T cells in spleen and lymph nodes were analyzed for Ki67 expression, a marker for proliferation. Representative plots show the frequencies of Ki67-positive control (CD45.1$^+$) and Cul4b$^{fl/fl}$-CD4Cre (CD45.2$^+$) CD4$^+$ T cells. The line graphs show the relative frequencies of Ki67$^+$CD4$^+$ T cells in spleen and lymph nodes. (H) The expression of activation markers CD69, CD25, and CD44 on control and Cul4b$^{fl/fl}$-CD4Cre CD4$^+$ T cells was analyzed by flow cytometry. The naïve CD4$^+$ T cells were stimulated with anti-CD3/CD28 mAb for 4 h and 24 h cells. The expression in naive and stimulated (4 and 24 h) control and Cul4b$^{fl/fl}$-CD4Cre CD4$^+$ T cells is shown. Black lines represent the control and red lines represent the Cul4b$^{fl/fl}$-CD4Cre CD4$^+$ T cells. For numerical raw data, please see S5 Data. Cul4b, Cullin-4b; mAb, monoclonal antibody.
(TIF)

**S5 Fig. Cul4b regulates cell cycle progression of activated CD4⁺ T cells. (A)** Cul4b-deleted (Cul4b^fl/fl^-CD4Cre) and control (Cul4b^fl/fl^) CD4⁺ T cells were cultured for 3 days and then rested in IL-2 for 2 days. After resting, these cells were restimulated for 4 h with anti-CD3/ CD28 mAbs (5 μg/ml). Proteins were quantified by iBAQ intensities values and were compared between control and Cul4b-deleted CD4⁺ T cells to generate fold changes. Volcano plot shows the differentially regulated proteins; blue dots indicate 517 proteins that were different ($P < 0.05$, $n = 3$). Proteins with higher abundance in CD4⁺ T cells derived from Cul4b^fl/fl^-CD4Cre are on the right side of the plot and those in control CD4⁺ T cells are on the left side. (**B**) CD4⁺ T cells were purified from control and Cul4b^fl/fl^-CD4Cre mice and stimulated with anti-CD3 and ant-CD28 (5 μg/ml) for 72 h. Cells were alcohol fixed and permeabilized and stained with PI, and DNA content was analyzed by flow cytometry. Doublets and dead cells were excluded. The histograms represent PI fluorescence intensity of CD4⁺ T cells. (**C**) The histogram shows the percentage of cells in G1, S, and G2/M phases of the cell cycle. Data are representative of 4 independent experiments. (*$P < 0.05$, by Student $t$ test). For numerical raw data, please see S6 Data. Cul4b, Cullin-4b; iBAQ, intensity-based absolute quantification; IL-2, interleukin 2; mAbs, monoclonal antibodies; PI, propidium iodide.
(TIF)

**S6 Fig. Expression profiling of CRL4B substrate receptors in CD4⁺ T cells. (A)** The list of known substrate receptors, their fold changes of protein abundance in Cul4b^fl/fl^-CD4Cre (KO) to control cells (WT), $P$ value, their respective copies, and concentration in control (WT) CD4⁺ T cells is shown. DCAF15, 10, 3, and 1 have log2 fold change of greater that 1 (= 2-fold change); among these DCAF1 has higher copy number and concentration. (**B**) Comparison of the transcript levels (by FPKM values) of substrate receptors listed in (A) in control (WT) and Cul4b^fl/fl^-CD4Cre CD4⁺ T cells. (**C**) The protein turnover of DCAF1 in anti-CD3 and anti-CD28 activated (for 24 h) control CD4⁺ T cells in the presence of translation inhibitor CHX was determined by immunoblotting; β-actin was used as loading control. For numerical raw data, please see S7 Data. For supporting data set, please see S1 and S2 Tables. CHX, cycloheximide; KO, knockout; WT, wild-type.
(TIF)

**S7 Fig. Cul4b interacts with Smc1a and drives its ubiquitination in activated CD4⁺ T cells. (A)** Coimmunoprecipitation of DCAF1, MRE11A, and SMC1A with Cul4b in TCR-stimulated CD4⁺ T cells. Protein lysates were preincubated with DNase I (10 μg/ml) for 30 min on ice prior to immunoprecipitation. (**B**) Coimmunoprecipitation of DCAF1 and Cul4b with SMC1A in TCR-stimulated CD4⁺ T cells. (**C–G**) Comparison of the transcript levels (by FPKM values) of proteins identified to be interacting with both Cul4b and DCAF1 in control (WT) and Cul4b^fl/fl^-CD4Cre CD4⁺ T cells. (**H**) Venn diagram shows the overlap of the proteins with increased diglycine-modified peptides (K-ε-GG) in control T cells and proteins identified as common interacting partners of Cul4b and DCAF1. Diglycine-modified peptides with 2-fold differences and proteins with at least 2 peptides IP'ed in mass spectrometry were used. (**I–K**) Comparison of the transcript levels (by FPKM values) of proapoptotic genes *Cdkn1a* (P21), *Bbc3* (Puma), and *Bax* in control (WT) and Cul4b^fl/fl^-CD4Cre CD4⁺ T cells. (**$P < 0.01$, by Student $t$ test). For numerical raw data, please see S7 Data. For supporting data set, please see S2 and S4 Tables. Cul4b, Cullin-4b; TCR, T cell receptor; WT, wild-type.
(TIF)

**S1 Table. Whole cell protein analysis with the up-regulated and down-regulated genes identified in Cul4b-deleted (KO) CD4⁺ T cells versus control (WT) CD4⁺ T cells.**
(XLSX)

**S2 Table. RNA-seq data showing differential gene expression in TCR-stimulated Cul4b-deleted (KO) CD4[+] T cells versus control (WT) CD4[+] T cells.**
(XLSX)

**S3 Table. The proteins preferentially immunoprecipitated by anti-Cul4b and anti-DCAF1 over anti-IgG in TCR-stimulated CD4[+] T cells identified by mass spectrometry.**
(XLSX)

**S4 Table. Di-glycine remnant enrichment analysis of differentially ubiquitinated proteins identified in control (WT) CD4[+] T cells versus Cul4b-deleted (KO) CD4[+] T cells.**
(XLSX)

**S1 Data. Numerical raw data.** All numerical raw data associated with Fig 1 and S1 Fig are combined in a single Excel file, "S1_Data." This file consists of several spreadsheets. Each spreadsheet contains the raw data of each subfigure.
(XLSX)

**S2 Data. Numerical raw data.** All numerical raw data associated with Fig 2 and S2 Fig are combined in a single Excel file, "S2_Data." This file consists of several spreadsheets. Each spreadsheet contains the raw data of each subfigure.
(XLSX)

**S3 Data. Numerical raw data.** All numerical raw data associated with Fig 3 and S3 Fig are combined in a single Excel file, "S3_Data." This file consists of several spreadsheets. Each spreadsheet contains the raw data of each subfigure
(XLSX)

**S4 Data. Numerical raw data.** All numerical raw data associated with Fig 4 are combined in a single Excel file, "S4_Data." This file consists of several spreadsheets. Each spreadsheet contains the raw data of each subfigure
(XLSX)

**S5 Data. Numerical raw data.** All numerical raw data associated with Fig 5 and S4 Fig are combined in a single Excel file, "S5_Data." This file consists of several spreadsheets. Each spreadsheet contains the raw data of each subfigure
(XLSX)

**S6 Data. Numerical raw data.** All numerical raw data associated with Fig 6 and S5 Fig are combined in a single Excel file, "S6_Data." This file consists of several spreadsheets. Each spreadsheet contains the raw data of each subfigure or multiple subfigures
(XLSX)

**S7 Data. Numerical raw data.** All numerical raw data associated with Fig 7, S6 and S7 Fig are combined in a single Excel file, "S7_Data." This file consists of several spreadsheets. Each spreadsheet contains the raw data of each subfigure or multiple subfigures
(XLSX)

**S1 Raw Images. All uncropped and original western blots images shown in main figures and supporting information.** Cells were washed with phosphate-buffered saline (PBS) ($Ca^{2+}$ and $Mg^{2+}$-free) and lysed using 1% Triton-X lysis buffer containing 50 mM Tris, 50 mM NaCl, Protease inhibitor cocktail (Roche), Halt phosphatase inhibitor cocktail (Thermo Fisher Scientific), $Zn^{2+}$-chelator *ortho-phenanthroline* (o-PA) (LifeSensors), deubiquitylase inhibitor PR-619 (LifeSensors), and 1% NP 40. Protein lysate was harvested and quantitated using Bradford reagent. For Cul4a and Cul4b neddylation experiments, whole-cell lysates were prepared in

SDS sample buffer (62.5 mM Tris-HCl, 2% w/v SDS, 10% glycerol, 50 mM DTT, and 0.1% bromophenol blue), vortexed to reduce sample viscosity, denatured by boiling, and then cooled on ice. Samples were resolved on 4%–12% Novex Tris-Glycine Gels (Thermo Fisher Scientific) and then transferred onto PVDF membrane (Amersham Pharmacia Biotech, Piscataway, New Jersey). The membrane was probed with the primary antibodies. Immunostaining was performed using appropriate secondary antibody at a dilution of 1:5,000 and developed with licor odyssey imaging system.
(PDF)

## Acknowledgments

We thank Natania Field, Patrick Lundgren, Rishabhadeva Tanga, and Kim Dale (The University of Pennsylvania) for their technical advice and support. We thank The Children's Hospital of Philadelphia flow cytometry core and animal facility for technical support. Part of this work was submitted to The American Association of Immunologists Annual Meeting, Immunology2020, and an abstract of the work was published in the Journal of Immunology [84].

## Author Contributions

**Conceptualization:** Asif A. Dar, Emma L. Lewis, Lynn A. Spruce, Paula M. Oliver.

**Data curation:** Asif A. Dar, Keisuke Sawada, Joseph M. Dybas, Emily K. Moser, Emma L. Lewis, Hossein Fazelinia, Lynn A. Spruce, Hua Ding, Steven H. Seeholzer.

**Formal analysis:** Asif A. Dar, Keisuke Sawada, Joseph M. Dybas, Emily K. Moser, Emma L. Lewis, Eddie Park, Hossein Fazelinia, Paula M. Oliver.

**Investigation:** Asif A. Dar, Paula M. Oliver.

**Methodology:** Asif A. Dar, Paula M. Oliver.

**Project administration:** Asif A. Dar.

**Supervision:** Asif A. Dar, Steven H. Seeholzer, Paula M. Oliver.

**Validation:** Asif A. Dar, Paula M. Oliver.

**Visualization:** Asif A. Dar, Emily K. Moser, Paula M. Oliver.

**Writing – original draft:** Asif A. Dar, Paula M. Oliver.

**Writing – review & editing:** Asif A. Dar, Keisuke Sawada, Joseph M. Dybas, Emily K. Moser, Emma L. Lewis, Eddie Park, Hossein Fazelinia, Lynn A. Spruce, Hua Ding, Steven H. Seeholzer, Paula M. Oliver.

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
