## [Editor Report · Decision Letter 0]

9 Mar 2020

Dear Dr Oliver, 

Thank you for submitting your manuscript entitled "Cul4b promotes CD4 T cell expansion by aiding the repair of damaged DNA" for consideration as a Research Article by PLOS Biology.

Apologies for the delay in getting back to you. Your manuscript has now been evaluated by the PLOS Biology editorial staff as well as by an academic editor with relevant expertise and I am writing to let you know that we would like to send your submission out for external peer review.

Please re-submit your manuscript within two working days, i.e. by Mar 11 2020 11:59PM.

Kind regards,

Di Jiang

PLOS Biology

---

## [Decision Letter · Decision Letter 1]

29 Apr 2020

Dear Dr Oliver,

I am very sorry for the delay in reviewing you manuscript "Cul4b promotes CD4 T cell expansion by aiding the repair of damaged DNA". We have now received comments from four reviewers and discussed them with the academic editor. 

In light of the reviews (below), we would welcome re-submission of a revised version that takes into account the reviewers' comments. As you will see, there are several experimental concerns that would seem to be required for the paper to be a good candidate for publication. Given the uncertain times we would be open to discussing a revision plan, and to being very flexible with revision times.

We cannot make any decision about publication until we have seen the revised manuscript and your response to the reviewers' comments. Your revised manuscript is also likely to be sent for further evaluation by the reviewers.

We expect to receive your revised manuscript within 2 months. 

**IMPORTANT - SUBMITTING YOUR REVISION**

*Re-submission Checklist*

*Published Peer Review*

*PLOS Data Policy*

*Blot and Gel Data Policy*

Sincerely,

Di Jiang

PLOS Biology

REVIEWS:

Reviewer #1: In the present manuscript, Dar et. al. investigate the role of an E3 ligase component, Cul4b, in CD4+/CD8+ T cell activation and expansion during an immune response following T cell receptor (TCR) activation. Transcription/translation of Cul4a and Cul4b were low in naïve T cells, but increased upon T cell activation with anti-CD3/28 antibodies ex vivo, whereas Cul4b seemed much more abundant than Cul4a as estimated by semi-quantitative mass spectrometry. Disruption of the Cul4b gene in murine CD4+ T cells through tissue-specific Cre-mediated LoxP recombination had no obvious effect on naïve CD4+/CD8+ T cells, but affected T cell expansion and activity upon TCR activation in Rag1-/- mice as a model system. Cul4b knock-out cells could still be activated through anti-CD3/28, but failed to expand due to impaired proliferation and higher rates of apoptosis. As the Cul4b knock-out cells also suffer an increase in DNA damage, the authors speculate that Cul4b is involved in the DNA damage response in these cells. Further experimentation identified the DCAF1 substrate receptor as the primary Cul4b-associated factor, suggesting that CRL4DCAF1 is the dominant CLR4 E3 ubiquitin ligase in these cells. Cul4b as well as DCAF1 immunoprecipitation experiments co-purified the Rad50, Mre11a and Smc1a, factors involved in DNA damage response. Depletion of Cul4b does not affect Smc1a recruitment to chromatin, but results in a Smc1a phosphorylation defect, a mark that is implicated in the activation of an S-phase checkpoint.

Overall, the performed experiments together with the obtained results support the majority of the conclusions that are drawn in this manuscript, and the Cul4b knock-out phenotype seems well characterized. As Cul4b is a known DNA damage repair (DDR) component, it is very likely that it carries out similar functions in activated CD4+/CD8+ T cells, although DDR may not be the only pathway that contributes to the observed phenotype. Cul4b is known to interact with more than 20 different DDB1-Cul4-associated factors (DCAFs), which also contribute to functions beyond DDR, e.g. cell cycle progression (i.e. through Cdt2). Disruption of Cdt2 activity is predicted to cause replication re-initiation during S-phase which by itself could lead to the activation of DNA damage checkpoints during S phase, proliferation defects and apoptosis. It is therefore not entirely clear whether increased DNA damage in these cells is (i) caused by the inactivation of CRL4-Cdt2, CRL4-DCAF1 or any other DCAFs, (ii) a consequence of a defective DNA damage response, or (iii) both. Providing the authors balance the interpretation and discussion of their findings in the light of these uncertainties, this reviewer would support publication in PLOS Biology. 

Specific comments:

Is it possible that the contribution of Cul4b to T cell proliferation might also be governed by other DDB1-Cul4-assocated factors (DCAFs) such as Cdt2, a CRL4 substrate receptor that controls entry into S phase through degradation of the replication licensing factor Cdt1?

Could DNA damage be a result of the Cul4b-/- cells failing to inactivate Cdt1 in S phase leading to replication re-initiation?

Western blots: How reproducible are the findings? To this reviewer it seems that claims of Figure 1a do not reproduce in Figure 2a. The relative abundance of unmodified versus neddylated Cul4b and Cul4a is not comparable between these two figures. 

In Figure 7B, DCAF1 levels markedly differ between WT and Cul4b KO cells after 24h stimulation, whereas in Figure 7C this difference is only detectable upon cycloheximide treatment, but not in the control. How would the authors explain this inconsistency? Also, does proteasome inhibition through MG132 restore DCAF1 levels in WT cells?

Figure 7G: IP westerns are not too convincing. For example, how reproducible is the co-immunoprecipitation of Smc1a? Smc1a seems hard to detect, but appears to be almost equally abundant in Suppl. Fig. 6E in the IgG control. Why are there two bands for DCAF1 in Fig. 7G, but only one in Figure 7B and C and Suppl. Fig. 6E? Can the authors be sure that the bands detected by their antibodies in Fig. 7G and Suppl. Fig. 6E indeed correspond to the appropriate target protein?

Minor comments:

p. 5, third paragraph: Figure SB probably refers to Figure S2B

p. 9, second paragraph: the authors state: '...given the abundance of pATM and g-H2AX staining in Cul4b-deficient T cells, it was clear that the MRN complex was not defective (Figure 7D-G). ...', but no data is not provided. 

Supplementary Fig. 6A: I think the authors mean -Log10(p value) in the second column, not +Log10(p value).

Reviewer #2: Manuscript ID: PBIOLOGY-D-20-00560R1

Title: Cul4b promotes CD4 T cell expansion by aiding the repair of damaged DNA

Authors: Dar et al.

The manuscript by Dar et al. builds on previous work by this laboratory discovering the upregulation of Cul4b in T cells following TCR activation to further describe the expression pattern of Cul4a and 4b as a function of T cell stimulation, the effect of conditional loss of Cul4b in CD4 T cells, the relative abundance of DCAF substrate receptors in the absence of Cul4b, and identification of potential interaction partners of Cul4b in T cells.

Overall, this is a solid body of work whose conclusions are, for the most part, supported by the experimental results. The finding that loss of Cul4b in T cells impairs T cell proliferation and survival without marked loss of cellularity, and is somewhat milder in phenotype than mice with conditional loss of DCAF1 in the T lineage is interesting, although the general phenotypes including defects in DNA repair, cell cycle progression, and increased apoptosis reported here have been observed in non-immune cell types, so they are not particularly surprising or novel. The identification of DCAF1 as a major Cul4b substrate interacting protein in CD4 T cells and the stabilization of DCAF1 upon loss of Cul4b provides strong evidence that DCAF1 serves as a major substrate receptor for Cul4b, but this work doesn't shed much light on what the cellular targets of the Cul4b-DDB1-DCAF1 E3 ligase might be. The attribution of greater DNA damage in Cul4b-deficient CD4 T cells as being due to its association with DDR factors is felt to be an overinterpretation, as the evidence for these interactions is not very convincing. Furthermore, the suggestion that Cul4b "preferentially associated with DCAF1" (pg. 4) may be somewhat misleading, as this conclusion is based mainly on pull-down assays that can be biased for abundant DCAFs rather than any systematic analysis of affinity for the large number of DCAFs identified as interacting with Cul4b. There were also some puzzling features of some of the experimental data that need fuller explanation. These and other concerns are described further below.

Major concerns:

1. It would be important for the authors to evaluate the T cell repertoire in the Cul4b conditional knockout mice, given evidence that loss of DCAF1 in B cells or T cells alters the immune repertoire (e.g. in B cells by dysregulating RAG1 levels and in T cells possibly interfering with DNA repair). This may or may not be masked by redundancy with Cul4a. This would also have implications for understanding the outcome of the adoptive transfer experiments in RAG1-/- mice.

2. There is a concern that undue emphasis was given to characterizing DCAF1, when levels of other DCAFs seem to be more strongly affected by loss of Cul4b. Furthermore, there is no biochemical evidence that DCAF1 "preferentially associates" with Cul4a, which could reflect a bias introduced by relative DCAF abundance.

3. A major weakness of the study is the relative lack of characterization of the transcript levels for the proteins identified as being up- or down-regulated by loss of Cul4b. The implicit assumption is that Cul4b is regulating protein turnover, but the effect could be indirect through transcriptional regulation.

4. The IP data suggesting Cul4b specifically associates with SMC1a and MRE11A is unconvincing (specificity/robustness in isotype control vs specific IP), poorly described, and needs further biochemical validation, including reciprocal co-IPs. In the absence of this, a direct link between Cul4b and DDR is deemed very tenuous, which may require revising this major conclusion and the title.

5. Figure 4b. The Cul4bfl/fl CD4Cre mice seem to show an accumulation of CD4lo CD3- T cells which is not discussed. Is it possible that Cul4b regulates CD3 expression?

6. There was a general absence of specific description of antibody reagents used for IP, western, and flow cytometry experiments (source, clone, etc). These should be included for transparency.

Minor concerns:

1. Figure 1. Both the neddylated and non-neddylated forms of Cul4a and Cul4b appear to increase in response to T cell activation. The graphs are plotted for each form, but this reviewer thinks it would be instructive to examine the ratio of the two forms.

2. Figure 5d. The presence of a fairly large number of CD45.1-CD45.2 double-positive cells in the lymph nodes is puzzling. Is this a gating artifact or non-specific staining?

3. Figure 6 D-J. The flow data should include specificity controls for intracellular staining, including isotype and fix-no perm controls for rigor and transparency.

4. The adoptive transfer of T cells to establish colitis in RAG1-/- mice is not particularly well justified, experimentally controlled, or interpreted. Why did the authors choose this approach rather than testing model T cell antigen-dependent immune responses? The experiment did not include a vehicle injected control, but only measures response against a positive control (Cul4bfl/fl T cells). Did the RAG1-/- animals receiving Cul4bfl/fl CD4Cre T cells show any evidence of inflammation compared to a negative control (i.e. vehicle injection)? The assertion of the authors that "Cul4b was required for the expansion and pathogenicity of activated CD4 T cells" (pg. 6) is an over-interpretation in the absence of negative control, and further data regarding immune repertoire diversity explained in Major Concern #1.

5. What is the source of the doublet in the DCAF1 western blots? The detection of one or two bands seems to vary under different conditions/in different experiments (e.g. Fig. 7C vs 7G).

6. Figure 7I purporting to show poor SMC1A activation (pSMC1A) in response to stimulation cannot be properly interpreted without a total SMC1A blot as a control. 

Reviewer #3: In this paper, the authors address a long-standing question on how T cell activation regulates the rapid cell proliferation. Following their previous proteomic studies that identified Cul4b being actively modified by neddylation, an indicative of functional activation, they focused on the role of Cul4b and its paralog Cul4a and their major substrate receptor, DCAF1, during T cell activations. They showed in this paper that during T cell activation, the steady state level and neddylated form of Cul4b was increased. Using a conditional Cul4b strain, they found that deletion of Cul4b impaired T cell proliferation and survival, and accumulated DNA damages. They further identified that Cul4b and its major binding partner, DCAF1, interact with multiple proteins involved in sensing and repair of DNA damage. SMC1A, a structural maintenance of chromosomes protein and a component of cohesin that is linked to the ATR/ATM DNA repair pathway, was implicated as a potential downstream factor of Cul4b. 

Overall, this is an important topic and most data they presented are clean. However, the paper, at its present form, is mere collection of some phenotypic descriptions and lack any mechanistic novelty. Multiple studies have demonstrated that the function of Cul4b, and DCAF1, is critical important for cell proliferation, survival and animal development. The role of Cul4b in DNA damage has been extensively characterized. The role of DCAF1 in T cell activation was also carefully characterized. Identifying Cul4b- and DCAF1-interacting proteins, as shown in Figure 6 and 7, is a good start, but the current study falls far short to provide novel insight without identifying the key or relevant protein(s) or substrate(s) that can explain the function of Cul4b in T cell activation. For example, is SMC1A a substrate of Cul4b or DCAF1? And is the regulation of DNA repair the main function of Cul4b during T cell expansion? Cul4b-deficient T cells apparently suffer significant defects during activation even in the absence of etoposide. 

1. Fig.5 claimed that "Cul4b promotes the maintenance of CD4 effector (CD44hiCD62Llo) T cell numbers" from the BM adoptive transfer model. However, this observation may be caused by a higher proliferation rate of control activated T cells when compared to Cul4b deficient cells.

2. Fig. 7 concluded Cul4b-DCAF1 complex in activated T cells may be involved in protein degradation as can be seen by higher DCAF1 level in Cul4b deficient cells but later they demonstrated SMC1A was phosphorylated rather than degraded by Cul4b-DCAF1 complex. Is the SMCA1 phosphorylation related to Cul4b or DCAF1 in any way? How does Cul4b deletion impairs SMC1A phosphorylation after exposure to etoposide? 

3. In the beginning, Fig. 1, the authors show Cul4b is upregulated and becomes active after activation of both CD4 and CD8 T cells but for the rest of the study they chose to merely focus on CD4 T cells. Does Cul4b play a similar role in CD8 T cells?

Reviewer #4: This manuscript by Dar et al. provided convincing evidence for a role of Cul4b in promoting T cell survival and expansion, and participating in DNA damage repair during T cell activation. In going forward, the authors may consider to address the following issues:

Major issues:

1. The authors claimed that in the absence of Cul4b, fewer cells entered the S and G2-M phase (Figure S5B-C) while more cells were undergoing apoptosis. It is not immediately obvious to this reviewer whether Cul4b-deficient cells had difficulty entering S phase compared with that of control. It would be helpful if Fig. S5C is presented with gated live cells so we can have a better picture of % of G1, S and G2/M between the two groups.

2. It is unclear what roles CUL4BDCAF1 plays in Smc1a phosphorylation. Is Smc1a ubiquitinated by Cul4b that facilitates ATM-dependent Smc1a phosphorylation? Are MRE11 and Rad50 ubiquitinated by Cul4b?

3. Cul4b is on X chromosome. The Cul4bfl/fl should be changed to Cul4bfl/Y.

Minor points:

1. Based on the results from cell transfer-induced colitis model and in vitro co-culture assay, CD4+ T cells require Cul4b to maintain survival and proliferation upon activation. it would be interesting to check under steady state, whether there is already altered/reduced CD4+ T cell number in colon of Cul4b deficient mice comparing to control mice, given that CD4+ T cells are mostly activated in intestine where large amount of commensal microbes colonize. 

2. How about the CD4+ T cell subsets (Th17, Th2, Treg) in colon between Cul4b deficient and control mice? 

3. In both colitis model and BM-chimera model, the in vivo proliferation (Ki-67, EdU) and apoptosis (active Caspase-3, Annexin V) of CD4+ T cells from Cul4b deficient donor and control donor should be determined.

4. In Fig. 2B, migration is not a reliable criteria to indicate that CBF Cul4b is mostly neddylated. it is possible that the CBF Cul4b migrates differently on SDS-PAGE due to high salt buffer used in the extraction. The authors may consider to compare Cul4b with or without MLN4924 treatment.

5. In Fig. 6B-C, a critical control was missing: Comet assay should be carried out in control and Cul4b deficient cells in the absence of Etoposide treatment.

6. In Fig. 7C-D, the authors should measure the half-life of DCAF1 post CHX treatment.

7. In Fig. S2B, CD3/CD28 or TCR stimulation increased CUL4b transcript. What about Cul4a?

8. In the main text and figure legends, please change all “CD4 T cells” to “CD4+ T cells” to be consistent with the labels in figures. Also, the “Rag-/-” should be “Rag-/-”.

9. On page 5, 7th line to the last, it should be “… of total Cul4 proteins in CD4+ T cells” as the assay was conducted on CD4+ T cells but not total T cells.

10. On page 5, 8th line to the last, it should be “(Figure 2C, Figure S2B)”.

---

## [Decision Letter · Decision Letter 2]

5 Oct 2020

Dear Paula,

Thank you very much for submitting a revised version of your manuscript "Cul4b promotes CD4 T cell expansion by aiding the repair of damaged DNA" for consideration as a Research Article at PLOS Biology. Thank you also for your patience as we completed our editorial process, and please accept again my apologies for the delay in providing you with our decision. This revised version of your manuscript has been evaluated by the PLOS Biology editors, the Academic Editor and two of the original reviewers.

The reviews are attached below. You will see that both reviewers find the manuscript very much improved, however they still raise several points that need to be addressed and also some statements that have to be toned down. Thus we are pleased to offer you the opportunity to address the points raised by the reviewers in a revised version that we anticipate should not take you very long. We will then assess your revised manuscript and your response to the reviewers' comments and we may consult the reviewers again.

We expect to receive your revised manuscript within 1 month.

**IMPORTANT - SUBMITTING YOUR REVISION**

*Resubmission Checklist*

*Published Peer Review*

*PLOS Data Policy*

*Blot and Gel Data Policy*

Best wishes,

Ines

--

Ines Alvarez-Garcia, PhD,

Senior Editor,

ialvarez-garcia@plos.org,

PLOS Biology

Reviewers’ comments

Rev. 1:

The authors state in the rebuttal: "While analyzing for the possible substrate receptors (SRs) which could explain the phenotype, we took multiple things into consideration, for example concentration of substrate receptors, fold change in the concentration of the SRs between control and Cul4bcKO cells, and extent to which each substrate receptor was immunoprecipitated with Cul4b (based on peptide count)"

Reviewer response: It is important to note that CRL4 substrate receptor levels (or fold-change upon CUL4b loss) do not - in any way - imply that the most abundant receptor is responsible for the process studied. Even low-abundance receptor levels can drive essential cellular processes. The linkage from CUL4b to DCAF1 to SMC1a, working together "to allow survival and expansion of activated T cells" is possible, yet this remains somewhat tentative with the data presented. See also comments by reviewer #2.

The authors state in the rebuttal: "Both buffer composition and incubation time greatly influences the extent of inhibition of the COP9 signalosome and thus neddylation of Cul4."

Reviewer response: I am not aware of this being formally shown in the literature. What is this statement based on? The CSN5 subunit of the signalosome is a Zinc protease, so Zn-chelators in the buffer could play a role in modifying activity. Good inhibitors are at hand for CSN though and commercially available (see https://www.nature.com/articles/ncomms13166). Could this comment possibly be an indication that the experimental read-outs are not very robust? See my previous comments (repeated below).

Previous comment: "Figure 7G: IP westerns are not too convincing. For example, how reproducible is the co- immunoprecipitation of Smc1a? Smc1a seems hard to detect, but appears to be almost equally abundant in Suppl. Fig. 6E in the IgG control. Why are there two bands for DCAF1 in Fig. 7G, but only one in Figure 7B and C and Suppl. Fig. 6E? Can the authors be sure that the bands detected by their antibodies in Fig. 7G and Suppl. Fig. 6E indeed correspond to the appropriate target protein?"

While this has been partially addressed by IP-MS for SMC1, which is appreciated, the identity/variability of the two vs. one band for DCAF1 remains a concern.

Rev. 2:

The resubmitted manuscript by Dar et al. has been substantially improved and has addressed most of the comments and concerns raised by the reviewers.

There remains some concern over the interpretation of the results and potential confounding variables, as well as a few points of clarification. These and other concerns are described further below.

Major concerns:

1. The authors justify not analyzing T cell repertoire by stating that "The rearrangement of TCR genes takes place before T cells enter the DP stage prior to Cul4b deletion." That is factually incorrect. TCR alpha chain rearrangement occurs during the DP stage. Thus, the original concern that the T cell repertoire may be significantly altered in the Cul4b conditional knockout mice, given evidence that loss of DCAF1 in B cells or T cells alters the immune repertoire (e.g. in B cells by dysregulating RAG1 levels and in T cells possibly interfering with DNA repair), still stands. This may or may not be masked by redundancy with Cul4a. This would also have implications for understanding the outcome of the adoptive transfer experiments in RAG1-/- mice, particularly if the TCR repertoire was severely restricted. TCR spectratyping analysis is quite feasible now and should answer the question.

2. The inclusion of the ratio of neddylated and non-neddylated forms of Cul4b in Fig. 1 confirms the suspicion that this ratio doesn't change much in response to TCR stimulation (none of the changes appears statistically significant). Thus, while the level of Cul4b increases robustly and is not in question, the activation of Cul4b is not necessarily enhanced. Thus, the authors should revise the some of the findings in the text and summary. For example, in the summary it states "Cul4b levels and activity increase following TCR stimulation", but the latter claim is not really supported by the analysis of Cul4b neddylation.

3. While the inclusion of additional data suggesting an association between Cul4b and SMC1a and MRE11A, and the reduction of SMC1a phosphorylation upon loss of Cul4b, is gratifying and improves rigor, the mechanistic link is at best only correlative. However, it is unclear what phospho-specific antibody is being used to detect SMC1. The antibody used is listed as being from Abcam, but there are several available for different sites: S360, S957, and S966. The specific product number is not included. The question is raised as to whether the phospho-site they are investigating makes sense with respect to TCR signaling, and loss of phosphorylation is a general phenomenon, or is site-specific. There is no justification provided for the choice of phospho-specific SMC1 antibody. This should be included at least, and would be more convincing if it were observed with another antibody as well (there are some available from CST which are better characterized).

4. In the previous version of the manuscript, the primary data for the mass spectrometry results was unavailable. It is interesting to note that the Cul4b IP, but not the VprBP/DCAF1 IP, pulls down another E3 ubiquitin ligase, Rbbp6, with as much confidence as other factors listed in Fig. 7F (12 peptides). Why this is noteworthy is that Miotto et al. published in Cell Reports (2014 Apr 24;7(2):575-587) that this E3 ligase plays a role in DNA replication and that its loss lead to increased spontaneous DNA damage, similar to the phenotype that is observed in the Cul4b-deficient cells. It is unclear why the identification of this protein was not noted in Fig. 7F. What the Miotto paper suggests is the possibility that Cul4b has a mechanistic link to Rbbp6. This should at least be acknowledged.

5. In re-reviewing the data in Fig. 6, it is not clear whether the authors can really attribute a role for Cul4b in the DNA damage response, because basal levels of activation of DDR factors are increased during activation, as the authors point out in line 294 "…Cul4b promotes the DNA damage response (DDR) in activated CD4+ T cells at a time point that correlates with DNA synthesis and proliferation". However, an alternative and nuanced interpretation is that due to Cul4b-dependent defects in DNA replication, spontaneous DNA damage increases, thus activating the DDR. Though levels of DDR factor staining in Fig. 6B increase upon etoposide treatment, it is not clear whether this is simply additive to (or significantly enhanced over) the effects observed by T cell stimulation. It is therefore felt that attributing a unique role of Cul4b in activating the DDR is not strongly supported by the evidence, and the text and title should be revised to reflect that possibility.

Minor concerns:

1. The authors should indicate how many technical replicates were performed for the mass spectrometry experiments (e.g. Fig. 7F).

---

## [Editor Report · Decision Letter 3]

20 Nov 2020

Dear Paula,

Thank you for submitting your revised Research Article entitled "Cul4b promotes CD4 T cell expansion by aiding the repair of damaged DNA" for publication in PLOS Biology. I have now obtained advice from the Academic Editor and have discussed the comments with the editorial team.

We're delighted to let you know that we're now editorially satisfied with your manuscript. We have a small suggestion to improve the title: "The ubiquitin ligase Cul4b promotes CD4+ T-cell expansion by aiding the repair of damaged DNA"

Before we can formally accept your paper and consider it "in press", we also need to ensure that your article conforms to our guidelines. A member of our team will be in touch shortly with a set of requests. As we can't proceed until these requirements are met, your swift response will help prevent delays to publication. Please also make sure to address the data and other policy-related requests noted at the end of this email.

- a cover letter that should detail your responses to any editorial requests, if applicable

*Copyediting*

*Published Peer Review History*

*Early Version*

Best wishes,

Ines

--

Ines Alvarez-Garcia, PhD,

Senior Editor,

PLOS Biology

ETHICS STATEMENT:

-- Please include the full name of the IACUC/ethics committee that reviewed and approved the animal care and use protocol/permit/project license. Please also include an approval number.

DATA POLICY:

Fig. 1B-F; Fig. 2C; Fig. 3C-G; Fig. 4A, B, C, E, F; Fig. 5A-L; Fig. 6C-K; Fig. 7D, H; Fig. S1A-C; Fig. S2A-C; Fig. 3A-E; Fig. S4A-H; Fig. S5A-C; Fig. S6B and Fig. S7C, D, E, F, G, I, J, K

For figures containing FACS data, we ask that you provide the FCS files.

Please also make the proteomics data deposited in Proteome Xchange publicly available.

---

## [Editor Report · Decision Letter 4]

15 Jan 2021

Dear Dr. Oliver,

I am writing concerning your manuscript submitted to PLOS Biology, entitled “The E3 ubiquitin ligase Cul4b promotes CD4+ T cell expansion by aiding the repair of damaged DNA.”

We have now completed our final technical checks and have approved your submission for publication. You will shortly receive a letter of formal acceptance from the editor.

Kind regards,

PLOS Biology